# Technical note: Fundamental aspects of ice nucleation via pore condensation and freezing including Laplace pressure and growth into macroscopic ice

Claudia Marcolli

Institute for Atmospheric and Climate Science, ETH, Zurich, Switzerland

*Correspondence to:* C. Marcolli (claudia.marcolli@env.ethz.ch)

**Abstract.** Pore condensation and freezing (PCF) is an ice nucleation mechanism that explains ice formation at low ice supersaturation. It assumes that liquid water condenses in pores of solid aerosol particles below water saturation, as described by the Kelvin equation, followed by homogeneous ice nucleation when temperatures are below about 235 K or immersion freezing at higher temperatures, in case the pores contain active sites that induce ice nucleation. Pore water is under tension (negative pressure) below water saturation as described by the Young-Laplace equation. This negative pressure affects the ice nucleation rates and the stability of the pore ice. Here, pressure dependent parameterizations of classical nucleation theory are developed to quantify the increase of homogeneous ice nucleation rates as a function of tension and to assess the critical diameter of pores that is required to accommodate ice at negative pressures. Growth of ice out of the pore into a macroscopic ice crystal requires ice supersaturation. This supersaturation as a function of the pore opening width is derived, assuming that the ice phase first grows as a spherical cap on top of the pore opening before it starts to expand laterally on the particle surface into a macroscopic ice crystal.

## 1 Introduction

Cirrus are high-altitude ice clouds that influence the Earth's climate by reflecting incoming solar short-wave radiation and regulating long-wave emissions to space resulting in a net warming effect (Stephens et al., 1990; Lohmann et al., 2008; Kärcher, 2017; Matus and l'Ecuyer, 2017). They vary in optical thickness and vertical extent depending on the atmospheric conditions and their formation mechanism (Kärcher, 2017; Kienast-Sjögren et al., 2016). Cirrus may form as outflow from convective or frontal clouds or in-situ when rising air parcels humidify while cooling (Krämer et al., 2016; Hartmann et al., 2018). Below the homogeneous ice nucleation threshold (HNT) at about 235 K, they can form through homogeneous ice nucleation (IN) in diluting liquid aerosol particles at relatively high ice supersaturation along the homogeneous freezing line of solution droplets (Koop et al., 2000), or heterogeneously at lower ice supersatuation aided by ice nucleating particles (INPs), which may induce freezing through immersion nucleation when coated with water-soluble material (Kärcher and Lohmann, 2003; Kuebbeler et al., 2014). Marcolli (2014) proposed that, in the absence of a coating, the prevailing mechanism of ice formation below water saturation is pore condensation and freezing (PCF). In PCF water that condensed in porous features of solid particles, freezes, and grows out of pores to form ice crystals. Indeed, most solid aerosol particles exhibit irregular surfaces with porous features such as cavities, slits, trenches, steps, and interstices between aggregated particles where liquid water can condense by capillary condensation below water saturation as described by the Kelvin equation. For temperatures below the HNT, pore water freezes homogeneously and may evolve into a macroscopic ice crystal by depositional growth. An indication for PCF is a distinct, almost step-like increase in the ice fraction below water saturation for temperatures below the HNT as compared to temperatures above it (Welti et al., 2014; Marcolli, 2014; 2017a). Such a jump in IN activity cannot be explained by applying classical nucleation theory (CNT) to deposition nucleation, assuming ice nucleation by deposition of water vapour with no liquid phase involved (Welti et al., 2014; David et al., 2019a). A distinct increase in ice fraction below the HNT has been observed for different particle types with inherent porosity. These include clay minerals with slits and trenches at particle edges (Marcolli et

al., 2014; 2017a; Wagner et al., 2016; Wang et al., 2016; David et al., 2019a), mesoporous silica particles and zeolites (Wagner et al., 2016; David et al., 2019a; 2019b), soot particles consisting of aggregated primary particles (Wagner et al., 2016; Mahrt et al., 2018; 2019; Nichman et al., 2019), porous glassy and crystallized particles (Wagner et al., 2012; 2014; Adler et al., 2013), and coal fly ash particles (Umo et al., 2019).

5    The role of pores for ice formation below water saturation has further been confirmed in microscopy studies of ice nucleation as ice crystals always formed at steps and imperfections where water may condense (Zettlemoyer et al., 1961; Wang et al., 2016; Kiselev et al., 2016; Pach and Verdaguer, 2019). Moreover, PCF has been directly observed for organic and water vapour condensing in wedge-shaped pockets on mica surfaces followed by crystallization and growth out of the confinement (Christenson, 2001; 2013; Kovács and Christenson, 2012; Kovács et al., 2012; Campbell et al., 2017; Campbell and Christenson, 10    2018).

The theoretical basis for PCF has been established already by Fukuta (1966), however, without the experimental data available to constrain the relevant conditions. Here, the different steps involved in PCF are analyzed, drawing from experimental data that has become available in the meantime. The conditions for pore filling, the stability of pore ice depending on temperature and pore width, and ice nucleation rates are derived, taking the effect of tension within pores into account. In addition to the 15   energy barrier associated with ice nucleation, previous studies have invoked a second energy barrier for ice growth out of pores (Page and Sear, 2006; Campbell and Christenson, 2018; Koop, 2017). Here, the conditions for an energy barrier-free ice growth out of pores as a function of pore opening diameter and ice supersaturation are derived. While the focus of most studies so far was on cylindrical pores, this technical note broadens the scope to trenches, wedges, and conical pores.

## 2 Atmospheric scenario of PCF

20   PCF can occur in pores of different geometries. Cylindrical pores and trenches fill completely at the relative humidity (RH) of pore filling. In case of pores with diameters of only few nanometers, filling and freezing even occurs below ice saturation. David et al. (2019a) have shown that growth of ice out of such narrow pores requires high ice supersaturation when they are isolated. Yet, when they are closely spaced, bridging of ice caps growing out of the pores greatly reduces the barrier for macroscopic ice growth. Conversely, conical and wedge-shaped pores combine the narrow bottom for water condensation and freezing below 25   ice saturation with a wide pore opening enabling ice growth out of the pore as soon as ice saturation is exceeded.

As an illustration of PCF in conical or wedge-shaped pores, Figure 1 depicts the atmospheric scenario of continuously increasing RH due to lifting of an air parcel. The pore surface is supposed to be wettable by liquid water but exhibits no ice nucleation activity. At low RH, a water layer forms on the pore surface and some liquid water condenses at the bottom of the pore (step 1 in Fig. 1). The pore water remains liquid as its volume is too small to host a critical ice embryo. When RH increases, more water 30   condenses (2) until the water volume becomes large enough to freeze. Because the water is under high tension, ice nucleation is expected to occur readily once the volume suffices to host the critical embryo (3). Ice nucleation is immediately followed by ice growth from the vapour phase (4), because at the same RH, ice is able to fill wider pores than water. As RH further increases, ice fills the pore more and more. Pores with narrow openings are completely filled well below ice saturation, while at ice saturation, pores of any widths are completely filled (5). Once ice saturation is exceeded, a spherical cap starts to grow on top 35   of the pore opening (6), and, when the angle of the cap has reached the critical value, further ice growth is unrestricted (7), and the typical ice habit develops (8). In the following, each of these steps will be analysed in detail and parameterizations to calculate pore filling, ice nucleation, and ice growth out of the pores will be given.

## 3 Capillary condensation in pores

In the atmosphere, pores of aerosol particles fill and empty in response to changes in ambient RH as the air cools and warms. 40   The Kelvin equation describes the equilibrium vapour pressure over curved surfaces and can be used to calculate RH of pore

filling by capillary condensation. It relies on the Young-Laplace equation that quantifies the pressure in liquids with curved surfaces (see Appendix A2 for a derivation of the Kelvin equation). Capillary condensed water forms a meniscus within pores as illustrated in Fig. 2 (panels c – e). Solving the Kelvin equation for the radius of the meniscus of a cylindrical or conical pore, $r_m(T)$, yields:

$$r_m(T) = \frac{2\gamma_{vw}(T)v_w(T,P_0)}{kT ln\frac{p}{p_w(T,P_0)}}.$$    (1)

Here, $\gamma_{vw}(T)$ is the surface tension of the vapour/water interface, $v_w(T,P_0)$ is the molecular volume of liquid water at standard pressure ($P_0 = 0.1$ MPa), $k$ is the Boltzmann constant, and $T$ is the absolute temperature. Finally, $S_w = \frac{p}{p_w(T,P_0)}$ denotes the saturation ratio, with $p_w(T,P_0)$ being the equilibrium vapour pressure above the flat water surface and $p$ the one above the curved surface. From Eq. 1 it becomes clear that the meniscus curvature increases with decreasing RH. Moreover, the radius $r_m(T)$

takes negative values for $S_w < 1$, indicating a concave curvature of the meniscus.

For wedge-shaped pores or trenches, the pressure difference across the interface between the vapour and the liquid phase is described by the Young-Laplace equation in its general form (Eq. A7) with two principal axes of curvatures, $r_1$ and $r_2$. As illustrated in Fig. 2b, $r_2$, the radius of curvature along the trench or wedge is assumed infinite ($r_2 = \infty$). Therefore, the Kelvin effect just depends on the radius of curvature $r_1$. As a function of the saturation ratio $S_w = p/p_w(T,P_0)$, it takes the form

$$r_1(T) = \frac{\gamma_{vw}(T)v_w(T,P_0)}{kT ln\frac{p}{p_w(T,P_0)}}.$$    (2)

The temperature dependence of the surface tension of liquid water can be calculated using the IAPWS (International Association for the Properties of Water and Steam) parameterization (Hrubý et al., 2014; Vinš et al., 2015):

$$\gamma_{vw}(T) = B\tau^{\mu}(1 + b\tau).$$    (3)

Here $\tau = 1 - T/T_c$ is the dimensionless distance from the critical temperature $T_c = 647.096$ K, $\mu = 1.256$ is a universal critical

exponent, coefficients $B$ and $b$ have values of 0.2358 Nm$^{-1}$ and -0.625, respectively, and $\gamma_{vw}(T)$ is given in units of J/m$^2$.

The dependence of the surface tension on curvature and pressure as described by the Tolman length, $\delta$, is believed to become relevant for strong curvatures (Schmelzer et al., 1996; Kalová and Mareš, 2015). Yet, the Tolman length for water and its temperature dependence are still debated. Recently, Kim et al. (2018) determined experimentally the Tolman length to be $\delta = 0.21 \pm 0.05$ nm, suggesting that the curvature dependence of the surface tension becomes relevant for pore diameters below 3

nm. However, modeling studies yield large discrepancies in its magnitude and sign (Malek et al., 2019), such that it is refrained here from implementing it in a parameterization.

The molecular volume of water depends on temperature and pressure. However, the Kelvin equation in its classical form makes the simplification to neglect the pressure dependence of the molecular volume (see Appendix A2). For consistency, the parameterization of the molecular volume for use in Eq. 1 includes temperature dependence at standard pressure ($P_0 = 0.1$ MPa)

and does not include a pressure dependent density:

$$v_w(T,P_0) = \frac{M_w}{N_a\rho_w(T,P_0)},$$    (4)

where $M_w$ is the molecular mass of water, $N_a$ is the Avogadro constant, and $\rho_w(T,P_0)$ is the temperature dependent density at standard pressure as parameterized in Eq. A1.

Conical pores are filled up to the diameter $D_p$ (see Fig. 2), which equals:

$$D_p = -2r_m(T)cos\theta_{ws},$$    (5)

where $\theta_{ws}$ is the contact angle of water on the pore surface, influencing the condensation of water and ultimately ice formation via PCF (Fukuta, 1966, David et al., 2019b). Cylindrical pores completely fill at the critical saturation ratio $S_c$ when Eq. 5 is fulfilled. In case of perfect wetting ($\theta_{ws} = 0°$), the pore radius corresponds with the radius of the water meniscus. As the ambient humidity increases above the value of complete pore filling, the curvature of the meniscus decreases (i.e. the curvature radius increases as shown in Fig. 2, panels c – e) and reaches infinity at water saturation.

According to the Young-Laplace equation, which describes the pressure difference $\Delta P$ across the vapour/water interface, a concave meniscus at the pore opening implies a negative pressure of the water within the pore, yielding for conical or cylindrical pores:

$$\Delta P = P - P_0 = \frac{2\gamma_{vw}(T)}{r_m(T)}, \tag{6}$$

where $P$ is the curvature dependent pressure within the pore water and $P_0$ is the standard pressure (0.1 MPa). For atmospheric applications, this pressure difference can be expressed as a function of the water saturation ratio, $S_w = p/p_w(T,P_0)$, as:

$$\Delta P = \frac{kT \ln \frac{p}{p_w(T,P_0)}}{v_w(T,P)}. \tag{7}$$

Thus, saturation ratios $S_w < 1$ yield negative pressure for water within the pore.

The Kelvin equation given in Eq. 1 assumes that the molecular volume of liquid water keeps the value at standard pressure ($v_w(T,P_0)$), which implies incompressibility (i.e. $\kappa(T) = 0$; see Appendix A2). Assuming constant compressibility of water instead, the pressure dependence of the molecular volume is taken into account so that Eq. 1 becomes (reformulating Eq. A17 of Appendix A2):

$$r_m(T,P) = \frac{\gamma_{vw}(T)\left(v_w(T,P) + v_w(T,P_0)\right)}{kT \ln \frac{p}{p_w(T,P_0)}}. \tag{8}$$

A parameterization of the pressure dependent molecular volume $v_w(T,P)$ is given in Eqs. A1 – A5.

The Laplace pressure of water within wedges and trenches can be formulated as:

$$\Delta P = \frac{kT \ln \frac{p}{p_w(T,P_0)}}{2 v_w(T,P)}. \tag{9}$$

When including the pressure dependence of the molecular volume for the curvature of the meniscus in trenches and wedges Eq. 2 becomes:

$$r_1(T,P) = \frac{\gamma_{vw}(T)\left(v_w(T,P) + v_w(T,P_0)\right)}{2kT \ln \frac{p}{p_w(T,P_0)}}. \tag{10}$$

Figure 3 illustrates the saturation ratio, $S_w$, above concave water surfaces (top panels) and the Laplace pressure within the liquid as a function of the meniscus curvature (bottom panels) for two temperatures. The pressure within the liquid is calculated using Eqs. 7 and 9 for cylindrical and wedge-shaped pores, respectively. With increasing concave curvature ($r_m \to 0$), the pore water is under increasing tension. Conical and wedge-shaped pores fill gradually with water as $S_w$ increases. In case of cylindrical pores, capillary condensation occurs when the pore diameter equals $D_p = -2r_m(T,P)\cos\theta_{ws}$. Similarly, in case of trenches, pore filling occurs when the slit width equals $-r_1(T,P)\cos\theta_{ws}$. At the RH of pore filling, the tension within the pore water is at its critical value for bubble nucleation (Blander and Katz, 1975; Marcolli, 2017b) and decreases when RH increases until the tension vanishes at water saturation. Taking the pressure dependence of the molecular volume into account (using Eqs. 8 and 10) results in a shift of the saturation ratio that is negligible given the uncertainties in the parameterization and compared with the temperature dependence (see Fig. 3).

## 4 Freezing of pore water

### 4.1 Homogeneous ice nucleation in bulk water

Classical nucleation theory (CNT) formulates the Gibbs free energy to create ice from water as the sum of a volume term, accounting for the energy released when a water molecule becomes part of the ice phase, and a surface term, accounting for the energy needed to build up the interface between ice and water. To compensate the energy invested in the buildup of the interface, an ice embryo needs a critical size to become stable (e.g. Lohmann et al., 2016). Since the surface-to-volume ratio is least for a sphere, CNT assumes spherical morphology of the emerging ice phase (Fukuta, 1966). While spherical morphology may seem

inappropriate considering the distinct faces of ice crystals representing the lattice symmetry, the spherical shape of ice embryos evolving in molecular dynamics simulations supports this assumption (Zaragoza et al., 2015).

The Gibbs free energy to form a spherical ice cluster with radius $r$ within the liquid phase depends on temperature $T$ and the absolute pressure $P$:

$$\Delta G(T,P) = 4\pi r^2 \gamma_{iw}(T,P) + \frac{4\pi r^3}{3v_i(T,P)}\Delta\mu_{iw}. \tag{11}$$

Here, $\gamma_{iw}(T,P)$ is the interfacial tension between ice and water, $r$ is the radius of the emerging ice embryo, $v_i(T,P)$ is the molecular volume of water in the ice phase, $\Delta\mu_{iw} = \mu_i(T,P) - \mu_w(T,P)$ is the difference between the chemical potentials of ice and liquid water, respectively.

The critical radius $r_c(T,P)$ of an ice embryo is reached when growth and shrinkage both lead to a decrease of the Gibbs free energy and can be determined by setting $\delta\Delta G/\delta r = 0$:

$$r_c(T,P) = \frac{2\gamma_{iw}(T,P)v_i(T,P)}{-\Delta\mu_{iw}}. \tag{12}$$

Accordingly, the Gibbs free energy barrier of homogeneous ice formation within the supercooled liquid water phase is given by:

$$\Delta G_c(T,P) = \frac{16\pi\gamma_{iw}(T,P)^3 v_i(T,P)^2}{3(-\Delta\mu_{iw})^2}. \tag{13}$$

### 4.1.1 Standard pressure

At standard pressure ($P_0 = 0.1$ MPa), the chemical potentials of liquid water and ice as a function of temperature are given as

$$\mu_w(T,P_0) = \mu_w(T_0,P_0) + kT\ln(p_w(T,P_0)), \tag{14}$$

and

$$\mu_i(T,P_0) = \mu_i(T_0,P_0) + kT\ln(p_i(T,P_0)). \tag{15}$$

Here, $p_w(T,P_0)$ and $p_i(T,P_0)$ are the equilibrium vapour pressures of liquid water and ice, respectively. At standard pressure $\mu_i(T,P_0) = \mu_w(T,P_0)$ when $T_0 = 273.15$ K. For $T < 273.15$ K, the chemical potential decreases when ice forms:

$$\Delta\mu_{iw} = \mu_i(T,P_0) - \mu_w(T,P_0) = kT\ln(p_i(T,P_0) - kT\ln(p_w(T,P_0) = -kT\ln\left(\frac{p_w(T,P_0)}{p_i(T,P_0)}\right). \tag{16}$$

Thus, the change in Gibbs free energy upon freezing can be formulated as a function of the equilibrium vapour pressures of water and ice, yielding at standard pressure:

$$\Delta G(T,P_0) = 4\pi r^2\gamma_{iw}(T,P_0) - \frac{4\pi r^3}{3v_i(T,P_0)}kT\ln\left(\frac{p_w(T,P_0)}{p_i(T,P_0)}\right). \tag{17}$$

Parameterizations of the equilibrium vapour pressures over a flat surface of water and ice at standard pressure are given in Murphy and Koop (2005):

$$\ln(p_w(T,P_0)) \approx 54.842763 - \frac{6763.22}{T} - 4.210\ln(T) + 0.000367T + \tanh(0.0415(T-218.8))(53.878 - \frac{1331.22}{T} - 9.44523\ln(T) + 0.014025T), \tag{18}$$

and

$$\ln(p_{ih}(T,P_0)) = 9.550426 - \frac{5723.265}{T} + 3.53068\ln(T) - 0.00728332T. \tag{19}$$

Eq. 19 applies to hexagonal ice (for T > 110 K), which is the stable ice phase at standard pressure (Murphy and Koop, 2005). However, there is evidence that at low temperatures metastable stacking disordered ice nucleates with stacking sequences representative of cubic (ABCABC) and hexagonal ice (ABABAB) (Kuhs et al., 2012; Koop and Murray, 2016; Hudait and Molinero, 2016; Amaya et al., 2017). The transition from hexagonal to stacking disordered ice involves an enthalpy increase of $\Delta G_{h\to sd} = 155 \pm 30$ J/mol between 180 and 190 K (Shilling et al., 2006). Using this value to obtain the equilibrium vapour pressure of stacking disordered ice yields (Murray et al., 2010; Němec, 2013; Laksmono et al., 2015; Koop and Murray, 2016):

$$p_{sd}(T,P_0) = p_{ih}(T,P_0)\exp\left(\frac{\Delta G_{h\to sd}}{RT}\right). \tag{20}$$

The interfacial tension between supercooled liquid water and the emerging ice phase ($\gamma_{iw}(T,P_0)$) is a key parameter in CNT but poorly constrained by experiments (Ickes et al., 2015). Experimental values are limited to 273.15 K when hexagonal ice and liquid water are in thermodynamic equilibrium. Since water becomes more ice-like with decreasing temperature, $\gamma_{iw}(T,P_0)$ is expected to decrease. Parameterizations of CNT differ in the value of $\gamma_{iw}(T,P_0)$ at the melting temperature and its temperature dependence (see Ickes et al. (2015) and Appendix B).

### 4.1.2 The role of pressure

The stability and nucleation rate of ice both depend on pressure. Since water condensing within pores at $S_w < 1$ is under tension (negative pressure), the impact of pressure needs to be taken into account. Pressure affects the chemical potentials of liquid water and ice. The chemical potential of liquid water as a function of pressure $P$ can be formulated as (Němec, 2013):

$$\mu_w(T,P) = \mu_w(T,P_0) + (P - P_0)\frac{v_w(T,P) + v_w(T,P_0)}{2}, \tag{21}$$

where $\mu_w(T,P_0)$ is the chemical potential at standard pressure. The parameterizations of the molecular volume of liquid water at standard pressure, $v_w(T,P_0)$, and including pressure dependence, $v_w(T,P)$, are given in Appendix A1 as Eq. A1 and Eqs A1 – A5, respectively.

Since the volume of ice I, which includes hexagonal ($I_h$), cubic ($I_c$), and stacking-disordered ice ($I_{sd}$) hardly changes under pressure, the pressure dependence of $\mu_i(T,P)$ can be formulated as (Němec, 2013):

$$\mu_i(T,P) = \mu_i(T,P_0) + (P - P_0)v_i(T,P_0). \tag{22}$$

Thus, the chemical potential difference between ice I and water is given as:

$$\Delta\mu_{iw} = \mu_i(T,P) - \mu_w(T,P) = \mu_i(T,P_0) - \mu_w(T,P_0) + (P - P_0)v_i(T,P_0) - (P - P_0)\frac{v_w(T,P) + v_w(T,P_0)}{2}. \tag{23}$$

Inserting Eq. 16 yields:

$$\Delta\mu_{iw} = (P - P_0)v_i(T,P_0) - (P - P_0)\frac{v_w(T,P) + v_w(T,P_0)}{2} - kT\ln\left(\frac{p_w(T,P_0)}{p_i(T,P_0)}\right). \tag{24}$$

Inserting Eq. 24 into Eqs. 12 and 13 yields pressure dependent formulations of the critical radius and the Gibbs free energy barrier, respectively, for homogeneous nucleation of pore ice.

Setting the chemical potential difference in Eq. 24 to zero ($\Delta\mu_{iw} = 0$) provides the condition for the pressure dependent melting curve of ice, which can be evaluated using the parameterizations of $v_i(T,P_0)$ and $v_w(T,P)$ given in Appendix A1 and the equilibrium vapour pressure of hexagonal ice, $p_{ih}(T,P_0)$. The excellent agreement between this evaluation (solid blue line) and the measured melting point depression of hexagonal ice (blue symbols) shown in Fig. 4 confirms the validity of Eq. 24 and the parameterizations of the molecular volumes of ice and water given in Appendix A1. Since the molecular volume of ice I is larger than the one of liquid water, increasing pressure decreases the melting temperature and applying tension increases it. The calculated melting temperature reaches a maximum for $P \approx$ -170 MPa with $T \approx$ 279 K.

Along with the melting point depression, there is a freezing point depression that can be described as a shift of the melting curve by $\Delta P$ = 307 MPa to lower pressures as shown in Fig. 4 (Koop et al. 2000; Marcolli, 2017b). To describe this freezing point depression using CNT, Eq. 24 is inserted into the parameterization of ice nucleation rates to account for the dependence of the chemical potentials on absolute pressure. Since CNT parameterizations differ in their formulation of ice nucleation rates, two different parameterizations, namely the ones by Murray et al. (2010; hereafter referred to as Mr10 parameterization) and Ickes et al. (2015; hereafter referred to as Ick15 parameterization) are used here (see Appendix B for their descriptions). Figure 4 shows that inserting the pressure dependent formulation of $\Delta\mu_{iw}$ in these CNT parameterizations decreases the freezing temperatures with increasing pressure. However, the calculated decrease does not describe the experimental data correctly. This is expected since the chemical potentials are not the only pressure dependent quantities in the parameterization of ice nucleation rates. To achieve agreement with the measured freezing temperatures, the pressure dependence of the other parameters also needs to be considered. Namely, in the Ick15 parameterization, the diffusion-activation energy depends on water diffusivity,

which is a pressure sensitive parameter. Therefore, its parameterization needs to be extended to include pressure dependence. Moreover, the pressure dependent formulation of the interfacial tension is adjusted to obtain agreement with the experimental freezing data (see Appendix B1). For the Mr10 parameterization, the interfacial tension is extended to include pressure dependence and adjusted to obtain agreement with the experimental data, while the pressure dependence of all other parameters

is neglected (see Appendix B2).

## 4.2 Stability of ice within pores

Freezing of pore water may occur when ice grows into the pore from the outside or when ice nucleates within the pore. For ice to form in confinement, the dimensions need to be large enough to host the critical embryo. For mesoporous silica, experiments have revealed the existence of a quasi-liquid layer (QLL) between ice and the pore surface with thickness $t$ of 0.38 to 0.6 nm

(Schreiber et al., 2001; Jähnert et al., 2008; Marcolli, 2014; Morishige, 2018). In order to incorporate an embryo of critical radius, a cylindrical pore therefore needs a diameter $D_p = 2r_c(T,P) + 2t$ (see Fig. 5 for illustration). The presence of a QLL adjacent to the pore wall provides an interface similar to bulk water such that the interfacial tension between the QLL and the ice embryo can be assumed the same as between bulk water and ice.

An ice embryo of critical size is metastable since $\Delta G(T,P) \geq 0$. To become stable, it needs to grow further until $\Delta G(T,P) \leq 0$.

In case of spherical growth, $\Delta G(T,P) = 0$ is reached when the embryo has a radius $r_s$ of:

$$r_s(T,P) = \frac{3\gamma_{iw}(T,P)v_i(T,P)}{-\Delta\mu_{iw}}. \tag{25}$$

While ice is free to grow spherically in bulk water, growth in pores is constrained by the pore walls. In cylindrical pores, once a spherical embryo has reached the pore wall, its growth is limited to the direction along the pore axis. Assuming that the spherical ice embryo cannot grow into the QLL and that the interfacial tension between the QLL and the pore ice is the same as

between bulk water and ice, the Gibbs free energy barrier for growth within a cylindrical pore is minimized when the ice embryo continues to grow as a cylinder with spherical caps on both ends once it has reached the pore walls (see Fig. 5 for illustration). The Gibbs free energy for such growth is given as:

$$\Delta G(T,P) = \gamma_{iw}(T,P)(4\pi r^2 + 2\pi ar) + \frac{\Delta\mu_{iw}}{v_i(T,p)}\left(\pi ar^2 + \frac{4\pi}{3}r^3\right), \tag{26}$$

with $r = D_p/2 + t$ equalling the maximum dimension a spherical ice embryo can reach within the pore, and $a + r$ representing

the extension of the growing ice cylinder along the pore as depicted in Fig. 5.

In Fig. 6, $\Delta G(T,P)$ is shown as a function of $a + r$ for different pore widths using the Ick15 parameterization at 230 K and $P_0 = 0.1$ MPa. The black dashed line indicates $\Delta G(230\ K,P_0)$ for the growth of a cylindrical ice embryo, starting from a thin disk with $r_c = D_p/2 + t,$ for comparison. The constant positive value of $1.68 \cdot 10^{-19}$ J arises from the contribution of the two ends of the cylinder to the Gibbs free energy. If these were neglected, the negative Gibbs energy from the volume term would exactly

compensate the positive contribution from the surface of the cylinder mantle, resulting in $\Delta G(T,P) = 0$ J. The black solid line represents ice nucleation in bulk water and is calculated assuming growth as a sphere using Eq. 26 with $a = 0$. It shows a steep decrease of $\Delta G(230\ K,P_0)$ once the energy barrier of $1.12 \cdot 10^{-19}$ J at the critical embryo size of $r_c = 1.095$ nm is overcome and reaches $\Delta G(230\ K,P_0) = 0$ J for $r_s = 1.643$ nm. Thus, a spherical pore or cage needs to be clearly larger than the critical size to host ice permanently.

When the width of a cylindrical pore is just sufficient to host the critical embryo, i.e. $r = r_c = 1.095$ nm, the red line in Fig. 6 is obtained, by first increasing $r$ until $r = r_c$ and then increasing $a$ while keeping $r = r_c$ constant in Eq. 26. It shows a constant Gibbs free energy, which remains at the critical value of $\Delta G(230\ K,P_0) = 1.12 \cdot 10^{-19}$ J, which is well below the Gibbs free energy of an ice cylinder within the pore, indicating that rounded caps are energetically favored compared with flat ends. If the pore is slightly wider than the critical size, pore ice becomes stable. The green line assumes that the ice embryo has a spherical shape

of $r = 1.1$ nm when it reaches the QLL and then starts to grow in length as a cylinder with half spheres at its ends. The Gibbs free energy for this pore drops below zero at $a + r = 152$ nm, i.e. requesting a pore of at least 304 nm in length to become stable.

The blue line describes ice growth within a slightly wider pore, such that the ice embryo has grown to a sphere of $r = 1.2$ nm when it reaches the QLL. Since for this pore width, the emerging ice embryo has overcome the energy barrier clearly before it has reached the pore wall, $\Delta G(230\,K,P_0)$ continuously decreases during further growth and reaches negative values already for $a + r = 8$ nm. Hence, given that the critical radius to host a stable ice phase within a cylindrical pore is only slightly larger than the critical embryo size, it is a good approximation to take $D_p = 2r_c + 2t$ as the pore diameter requested to host ice stably within the pore. Note that a large uncertainty in this expression stems from $t$, the thickness of the QLL, which is difficult to measure and depends on temperature (Webber and Dore, 2004; Webber et al., 2007).

Molecular dynamics simulations have shown recently that for $r \approx r_c$ such that $\Delta G\,(T,P) > 0$, liquid water and ice coexist in time through oscillations between all-liquid and all-crystalline states (Kastelowitz and Molinero, 2018). When $r$ is slightly larger such that $\Delta G\,(T,P) < 0$, all bulk water is frozen.

Figure 7 shows that for the conditions used in Fig. 6 ($S_w = 1$; $P = 0.1$ MPa, and $T = 230$ K) the Mr10 parameterization predicts slightly larger critical radii of 1.24 nm for $n = 0.97$ and $r_c = 1.26$ nm for $n = 0.3$ compared to $r_c = 1.095$ nm for the Ick15 parameterization. This exemplifies the uncertainty in critical radius depending on CNT parameterization. These critical sizes are applicable to ice formation within pores of particles immersed in water (i.e. prepared as a slurry). The critical size for pore ice decreases with decreasing water saturation as shown in Fig. 7. When porous particles are exposed to air with $S_w < 1$, the water that condenses within the pores is under negative pressure and the critical radius requested to keep ice stable decreases. Thus, at $S_w = 0.3$, implying a pressure of -120 MPa within the pore water, the critical radius decreases to 0.73 nm for the Ick15 parameterization and to 0.84 nm for Mr10 with $n = 0.97$ and to 0.91 nm for Mr10 with $n = 0.3$. Thus, in very narrow pores, ice may be stable at low RH due to the negative pressure but melt when RH increases.

## 4.3 Homogeneous ice nucleation within pore water

Even if pores are large enough to host ice, pore water may remain liquid when ice nucleation rates are too low. Figure 8a shows the pressure dependence of homogeneous nucleation rates for the Ick15 and Mr10 parameterizations (with $n = 0.3$ and 0.97) at four different temperatures from 235 K to 210 K. In panel (b), the nucleation rates are converted to times needed to freeze a water volume corresponding to the critical embryo size. All parameterizations predict a strong increase of nucleation rates with negative pressure; however, they differ in the degree of this increase.

According to the Ick15 parameterization, the nucleation rate at 230 K at ice saturation (causing a tension of -42 MPa within the pore water) is $1.34 \cdot 10^{17}$ cm$^{-3}$s$^{-1}$. With this rate, it takes about 0.5 h for a critical water volume to freeze. Conversely, in a cylindrical pore of 3.3 nm width and 500 nm length implying a radius $r = 1.25$ nm available for free water (assuming $t \approx 0.4$ nm), freezing takes place within approximately 3 s. However, close to water saturation, when the pore water experiences ambient pressure, freezing of water in such a pore takes almost a day, highlighting the strong impact of pressure on ice nucleation. At 30 % RH$_w$, which corresponds with the pore filling RH for this pore width, the pore water, which is at -124 MPa, freezes within $6 \cdot 10^{-5}$ s, and even a critical water volume should freeze within ~0.1 s.

The Mr10 parameterization predicts similar trends, however, with a stronger increase of nucleation rates with decreasing temperature and increasing tension. Assuming ice saturation at 230 K, a critical water volume freezes in no more than 2 min for the $n = 0.3$ parameterization and in 5 s for $n = 0.97$, while the 500 nm long pore freezes within 0.3 s ($n = 0.3$) and 0.01 s ($n = 0.97$). At water saturation, the pore water takes half a day ($n = 0.3$) and about an hour ($n = 0.97$) to freeze, while at 30 % RH$_w$ even a critical water volume should freeze immediately (within $2 \cdot 10^{-5}$ s for $n = 0.3$ and $10^{-6}$ s for $n = 0.97$).

At lower temperatures (220 K and 210 K), the Ick15 parameterization predicts only a slight increase of nucleation rates with decreasing pressure, while both Mr10 parameterizations (with $n = 0.3$ and $n = 0.97$) predict higher rates than the Ick15 parameterization at ambient pressure and a stronger increase when water is under tension. These discrepancies between parameterizations reflect that homogeneous ice nucleation rates at standard pressure are not well constrained for temperatures below 230 K. Despite these discrepancies, all parameterizations agree that pore water is able to freeze within atmospherically

relevant timescales. This finding is indeed confirmed in freezing experiments performed with mesoporous silica particles with closely spaced cylindrical pores of 3.8 nm diameter, which grew into macroscopic ice crystals within about 10 seconds at 228 K but not at 233 K (David et al., 2019a). Moreover, all parameterizations predict increasing freezing rates with decreasing relative humidity, i.e. the higher the tension is within the pore water. Such a behavior was actually observed for mesoporous

silica particles with 9.1 nm pore diameter, which manifested decreasing ice fraction with increasing RH at 233 K (David et al., 2019b).

Above the HNT, homogeneous ice nucleation rates decline and nucleation sites on the pore wall are required to induce freezing of pore ice.  Thus, freezing needs to occur in immersion mode.

## 5 Ice growth from the vapour phase

Once a critical embryo forms within a pore, freezing consumes all pore water almost instantly and further ice growth needs to occur by water vapour deposition. Since cylindrical pores and trenches completely fill with water once the RH of pore filling is reached, ice nucleation leads to pores completely filled with ice. Conversely, conical pores and wedges gradually fill with water such that pores are only partly filled with ice at the instant of pore water freezing. Hence, in this case growth from the vapour phase starts already within the pores.

For growth out of the pore, the pore opening needs to be wide enough or pores need to be closely spaced (David et al., 2019a). In the following, the conditions for growth of ice within pores and out of pores are derived.

### 5.1 Ice growth within conical pores and wedges

Assuming that the Kelvin effect also applies to ice, the pore ice should form a concave meniscus at the ice/vapour interface to stabilize the ice phase with respect to evaporation below ice saturation (Fukuta, 1966). Such a curvature can be realized through

a curved QLL on top of the ice surface. Using the Kelvin equation to describe the equilibrium condition of ice with respect to vapour yields the following diameter of pore filling for conical pores:

$$D_p(T) = \frac{-4\gamma_{vi}(T)v_i(T,P_0)cos\theta_{is}(T)}{kTln\frac{p}{p_i(T,P_0)}}. \tag{27}$$

Here, $\gamma_{vi}(T)$ is the surface tension of ice, and $\theta_{is}(T)$ is the contact angle between ice and the pore surface. The ratio $p/p_i(T, P_0)$ yields $S_i$, the supersaturation with respect to ice.

In case of wedge-shaped pores, the diameter of pore filling is given as:

$$D_1(T) = \frac{-2\gamma_{vi}(T)v_i(T,P_0)cos\theta_{is}(T)}{kTln\frac{p}{p_i(T,P_0)}}. \tag{28}$$

The surface tension of ice is not well known. Yet, assuming that a QLL forms at the ice/vapour interface, an upper limit can be estimated as the sum of the surface tension of water and the interfacial tension between water and ice: $\gamma_{vi}(T) = \gamma_{vw}(T) + \gamma_{iw}(T, P_0)$ (David et al., 2019a).

Assuming that surface wetting precedes capillary condensation within the pore such that the whole pore surface is covered by adsorbed water when the pore water freezes, the contact angle $\theta_{is}(T)$ in Eqs. (27) and (28) can be replaced by the one between ice and water. Thus, the contact angle between ice and the substrate can be substituted by the one between ice and water:

$$cos\theta_{iw}(T) = \frac{\gamma_{vw}(T)-\gamma_{iw}(T,P_0)}{\gamma_{vi}(T)}. \tag{29}$$

Figure 9 compares pore filling with ice and water at 230 K, assuming that the growing ice phase is hexagonal and using $\gamma_{iw}(T)$

from the Ick15 parameterization and $\gamma_{vw}(T)$ as parameterized in Eq. 2. With these assumptions, a contact angle of $\theta_{iw}(230 \text{ K}) \approx$ 55° results. Since Fig. 3 showed that using pressure dependent molecular volumes has little impact on pore filling, this effect is neglected here. Pore filling is calculated once with the assumption that the adsorbed water layer is involved in the curvature of the meniscus, and once assuming that it is not involved, such that its presence narrows the effective pore diameter by 2*t*. Note

that the thickness of the adsorbed water layer and the width of the QLL between the pore surface and ice do not need to coincide. Yet, since both values are not well constrained we assume them the same. Figure 9 shows that for $S_w > 0.25$, ice is able to fill wider pores than liquid water. Moreover, the pore filling extends to larger diameters for conical pores (left panels) than for wedge-shaped pores (right panels), since conical pores are constrained in two dimensions and wedge-shaped pores only in one.

The width of the QLL is significant for narrow pores (upper panels) but loses its relevance for wide pores (lower panels). At ice saturation, the pore diameter for filling with ice diverges to infinity, while pore filling with liquid water is still restricted to narrow pores. Thus, at ice saturation, all pores fill with ice up to the pore opening, while liquid water remains restricted to the narrow bottom of conical and wedge-shaped pores.

## 5.2 Ice growth out of a pore

For an energy-barrier-free ice growth out of a pore, the energy cost to build-up additional surface needs to be balanced by the energy gain due to the increase in ice volume. To realize ice volume growth with minimal increase of ice surface area, ice is assumed to grow as a spherical cap as illustrated in Fig. 10. Assuming such growth, the energy balance is given as:

$$\Delta G_{gr}(T, P) = \pi \left( \left( r_{op} + x \right)^2 + h^2 - r_{op}{}^2 \right) \gamma_{vi}(T) + \pi \left( (r_{op} + x)^2 - r_{op}{}^2 \right) \gamma_{is}(T) - \frac{\pi h}{6 v_i(T, P_0)} \left( 3 \left( r_{op} + x \right)^2 + h^2 \right) kT \ln \frac{p}{p_i(T, P_0)} \quad .$$
(30)

Here, $\gamma_{is}(T)$ is the interfacial energy between ice and the outer surface surrounding the pore, $r_{op}$ is the radius of the pore opening, $h$ the height of the spherical cap and $x$ is the radius increase of the base of the spherical cap to the outer surface as shown in Fig. 10. The first term on the right side of the equation describes the energy increase due to the increase of the ice/vapour interface, the second one is the energy increase due to the increase of the interfacial area between ice and the outer particle surface, and the third is the energy decrease due to the increase of ice volume. When RH exceeds ice saturation, a spherical cap forms on

top of the pore opening. With increasing ice supersaturation $S_i = \frac{p}{p_{ih(T, P_0)}}$, it increases first in height $h$ without any extension of the cap base ($x = 0$) until the contact angle realizes the critical value for unlimited growth. For the assumption that the outer surface is covered with an adsorbed water layer, $\gamma_{is}(T)$ in Eq. 29 can be substituted by $\gamma_{iw}(T, P_0)$ and the contact angle for unlimited growth is given as the one between ice and water, $\theta_{iw}(T)$.

The assumption that ice needs to grow to a spherical cap with a contact angle $\theta_{iw}(T)$ yields for cylindrical and conical pores the

following pore opening for free growth:

$$D_{pfg}(T) = \frac{4\gamma_{vi}(T) v_i(T) \sin \theta_{iw}(T)}{kT \ln \frac{p}{p_{ih}(T, P_0)}},$$
(31)

and for wedge-shaped pores and trenches:

$$D_{1fg}(T) = \frac{2\gamma_{vi}(T) v_i(T) \sin \theta_{iw}(T)}{kT \ln \frac{p}{p_{ih}(T, P_0)}}.$$
(32)

Figure 11 shows that large ice supersaturations are needed for growth out of a narrow pore. At $S_i = 1.1$, conical pore openings

need to be 36 nm in diameter to allow unrestricted ice growth out of the pore, while for wedge-shaped pores 18 nm suffice.

A pore filled with ice can be viewed as a perfect active site for deposition nucleation with a contact angle of 0, such that there is no thermodynamic energy barrier for ice nucleation, i.e. $exp(\Delta G_c \theta_{is}/kT) = 1$ (see CNT formulation for heterogeneous ice nucleation in e.g. Zobrist et al., 2007, or Kaufmann et al., 2017). For contact angles larger than zero, there is an energy barrier and the active site size for growth into an ice crystal needs to be larger. In other words, deposition nucleation occurring on active

sites requires IN active areas that are larger than pore openings for unrestricted ice growth. Thus, the required area for deposition nucleation occurring on a flat surface needs to be much larger than the one required for immersion freezing as determined e.g. in Kaufmann et al. (2017). This makes it unlikely that immersion freezing sites are sufficiently large to host a critical embryo in deposition mode. Therefore, immersion mode active sites present on the flat particle surface should be irrelevant for deposition nucleation.

The Kelvin equation can also be used to calculate the diameter for unrestricted (energy-barrier-free) growth of a hypothetical spherical ice particle. Using the same parameters as for growth of pore ice, yields a diameter of 44 nm at $S_i$ = 1.1 and $T$ = 230 K (see Fig. 11). Smaller ice particles shrink due to sublimation. This large diameter for unrestricted ice growth arises from the high surface tension of ice. In the atmosphere, surface tensions may be lowered due to adsorption of semivolatile organic vapours. The dashed lines in Fig. 11, which were calculated with the interfacial tension $\gamma_{iw}(T)$, and the surface tensions $\gamma_{vw}(T)$, and $\gamma_{vi}(T)$ all halved, show that reduced interfacial tensions facilitate growth of ice. Trace amounts of semivolatile organic vapours can be assumed omnipresent and should influence PCF mainly through reducing surface tensions. The condensation of larger amounts of water-soluble organics in pores influence PCF through lowering the water activity as discussed in Marcolli (2017a). When solid particles have acquired a thick coating, pores become irrelevant and freezing may occur through immersion freezing for particles with nucleation sites or homogeneously along the homogeneous freezing line of solution droplets.

The supersaturation required for ice growth out of pores in Fig. 11 applies to single isolated pores. Using CNT and molecular dynamics simulations, David et al. (2019a) showed that a network of closely spaced pores lowers the supersaturation required for macroscopic ice-crystal growth out of narrow pore openings through bridging of ice caps growing out of adjacent pores.

**6 Conclusions**

The conditions derived for ice nucleation within pores and growth of ice out of pores show that porous particles are able to nucleate ice at low ice supersaturation and well below water saturation. The focus of this technical note is on homogeneous ice nucleation within pores, which occurs below the HNT. Above the HNT, ice nucleation needs to occur heterogeneously on nucleation sites within pores that are active in immersion mode. Such nucleation sites are considered specific for each aerosol particle type. Even though porosity can be considered as a surface characteristic, PCF below the HNT should not be viewed as a heterogeneous ice formation process, but as homogeneous freezing because the formation of the ice phase occurs within the volume of the supercooled pore water and not on the pore surface.

Well suited for ice formation by PCF are particles with conical and wedge-shaped pores, or with narrow pores that are closely spaced. Surface roughness ranging from the small to large nanometer scale is suitable for water condensation, freezing, and ice growth. This makes PCF the likely mechanism for ice formation at low ice supersaturation. Deposition nucleation on the other hand is unlikely if one considers the much larger IN active areas needed for deposition nucleation than for immersion freezing. Yet, the atmospheric relevance of PCF depends on the coating of the aerosol particles. Trace amounts may indeed promote ice growth out of pores, if they reduce the surface tension of ice. When solid particles have acquired a thick coating, pores likely become irrelevant. In these cases, ice formation may occur through immersion freezing for particles that act as INP or along the homogeneous freezing line of solution droplets.

**Appendix A**

**A1 Temperature and pressure dependent densities of supercooled liquid water and ice**

The equation of state relates density with the state variables temperature and pressure. Water shows a density maximum at 277.13 K at standard pressure that shifts to warmer temperatures for negative pressures (Pallares et al., 2016) and vanishes at high pressures (Mishima, 1996; Holten and Anisimov, 2012). Marcolli (2017a) proposed a parameterization of liquid water density at standard pressure (0.1 MPa) in units of kgm$^{-3}$ with a validity range from 50 to 393 K:

$\rho_w(T, P_0) = 1864.3535 - 72.5821489 \cdot T + 2.5194368 \cdot T^2 - 0.049000203 \cdot T^3 + 5.860253 \cdot 10^{-4} \cdot T^4$

$-4.5055151 \cdot 10^{-6} \cdot T^5 + 2.2616353 \cdot 10^{-8} \cdot T^6 - 7.3484974 \cdot 10^{-11} \cdot T^7 + 1.4862784 \cdot 10^{-13} \cdot T^8$ \hfill (A1)

$-1.6984748 \cdot 10^{-16} \cdot T^9 + 8.3699379 \cdot 10^{-20} \cdot T^{10}.$

To account for its pressure dependence, the density of liquid water can be formulated in terms of the compressibility $\kappa(T)$ and its derivative $\partial\kappa(T)/\partial P$:

$$\rho_w(T,P) = \rho_w(T,P_0) + \kappa(T) \cdot P + \partial\kappa(T)/\partial P \cdot P^2. \tag{A2}$$

Density data, covering the pressure and temperature ranges from 0.1 to 399 MPa and 200 – 300 K, respectively (Hare and
Sorensen; 1987; Mishima, 2010; Holten and Anisimov, 2012), together with density data from Pallares et al. (2016), covering the range from standard pressure to -110 MPa and temperatures from 258.15 to 333.15 K, were used to parameterize $\kappa(T)$ in units of MPa$^{-1}$:

$$\kappa(T) = 0.487 - 0.004368 \cdot (T - 273.15) + 0.00007235 \cdot (T - 273.15)^2, \tag{A3}$$

and $\partial\kappa(T)/\partial P$ in units of MPa$^{-2}$:

$$\partial\kappa(T)/\partial P = -0.0003805 + 6.639 \cdot 10^6 \cdot (T - 273.15) - 9.688 \cdot 10^8 \cdot (T - 273.15)^2 \ . \tag{A4}$$

Inserting Eqs. A3 and A4 in Eq. A2 yields a parameterization for the density of water in units of kgm$^{-3}$ that is valid from 203.15 to 333.15 K and -110 – 399 MPa with a standard deviation of 3 kg/m$^3$, and maximum deviations of $\pm$ 10 kg/m$^3$.

With this density, a temperature and pressure dependent molecular volume of liquid water can be formulated as:

$$v_w(T,P) = \frac{M_w}{N_a \rho_w(T,P)}. \tag{A5}$$

The temperature dependent molecular volume at standard pressure, $v_w(T,P_0)$, is obtained by inserting Eq. A1 into Eq. A5. The density of ice I is only slightly pressure dependent. Neglecting this pressure dependence, it can be parameterized as (Zobrist et al., 2007):

$$v_i(T,P) \approx v_i(T,P_0) = \frac{M_w}{N_a \rho_0}\left(1 - 0.05294\frac{T-273.15\,K}{273.15\,K} - 0.05637\left(\frac{T-273.15\,K}{273.15\,K}\right)^2 - 0.002913\left(\frac{T-273.15\,K}{273.15\,K}\right)^3\right)^{-1}. \tag{A6}$$

Using Eq. A6, the molecular volume of hexagonal ice is 3.264·10$^{-29}$ m$^3$ at 273.15 K and decreases to 3.231·10$^{-29}$ m$^3$ at 200 K.
The same density parameterization is used for hexagonal, cubic and stacking disordered ice because diffraction data showed that the densities of ice Ih and ice Ic are the same within experimental uncertainty (Murray et al., 2010; Dowell and Rinfret, 1960).

## A2 Derivation of the Kelvin equation

The Young-Laplace equation describes the pressure difference $\Delta P$ across an interface with interfacial tension $\gamma(T,P)$ as a
function of the curvature of the surface. In its general form, it is given as:

$$\Delta P = \gamma(T,P)\left(\frac{1}{r_1} + \frac{1}{r_2}\right), \tag{A7}$$

with $r_1$ and $r_2$ being the principal radii of curvature, which are orthogonal to each other.

In case of a curved water surface in contact with its vapour the Young-Laplace equation becomes:

$$\Delta P = P - P_0 = \gamma_{vw}(T,P)\left(\frac{1}{r_1} + \frac{1}{r_2}\right). \tag{A8}$$

with $P$ being the absolute pressure within the liquid and $P_0$ the pressure over the flat surface. For a sphere with $r_1 = r_2 = r_m$ one obtains Eq. 6 of the main text.

Thus, underneath convex surfaces, such as spherical cloud droplets, pressure is increased, whereas, underneath concave surfaces, such as the meniscus of capillary condensate/pore water, pressures are negative, which corresponds with a tension.

In thermodynamic equilibrium, the chemical potentials of water vapour and liquid water are equal:

$$\mu_v(T,P) = \mu_w(T,P). \tag{A9}$$

If additional pressure is exerted the chemical potential of the liquid phase changes to:

$$\mu_w(T,P) = \mu_w(T,P_0) + \int_{P_0}^{P} v_w(T,P')dP'. \tag{A10}$$

Integration yields:

$$\mu_w(T,P) = \mu_w(T,P_0) + (P - P_0)\frac{v_w(T,P)+v_w(T,P_0)}{2}. \tag{A11}$$

Here, the temperature and pressure dependent formulation of the molecular volume, $v_w(T,P)$, as parameterized in Appendix A1 can be used.

Similarly, the pressure dependence of the chemical potential of the vapour phase can be formulated as:

$$\mu_v(T,P) = \mu_v(T,P_0) + \int_{P_0}^{P} v_v(T,P')dP'. \tag{A12}$$

Using the ideal gas law, the molecular volume of the gas phase as a function of the vapour pressure $p$ is given as:

$$v_v(T,P) = \frac{kT}{p} \tag{A13}$$

Insertion into Eq. A12 and integration yields:

$$\mu_v(T,P) = \mu_v(T,P_0) + kT\ln\frac{p}{p_w(T,P_0)}. \tag{A14}$$

Inserting Eqs. A11 and A14 in Eq. A9 yields:

$$\mu_v(T,P_0) + kT\ln\frac{p}{p_w(T,P_0)} = \mu_w(T,P_0) + (P - P_0)\frac{v_w(T,P)+v_w(T,P_0)}{2}. \tag{A15}$$

Since in thermodynamic equilibrium $\mu_v(T,P_0) = \mu_w(T,P_0)$, Eq. 15 simplifies to:

$$kT\ln\frac{p}{p_w(T,P_0)} = (P - P_0)\frac{v_w(T,P)+v_w(T,P_0)}{2}. \tag{A16}$$

Using the Young-Laplace equation as given in Eq. 6 to calculate the pressure change $(\Delta P = P - P_0)$ due to the curvature of the water surface yields:

$$\ln\frac{p}{p_w(T,P_0)} = \left(\frac{2\gamma_{vw}(T)}{r_m kT}\right)\frac{v_w(T,P)+v_w(T,P_0)}{2} \tag{A17}$$

The Kelvin equation results when the liquid phase is assumed incompressible ($v_w(T,P) = v_w(T,P_0)$):

$$\ln\frac{p}{p_w(T,P_0)} = \frac{2\gamma_{vw}(T)v_w(T,P_0)}{r_m kT}. \tag{A18}$$

## Appendix B: Parameterizations of ice nucleation rates

CNT describes ice nucleation as an activated process with a thermodynamic energy barrier and a pre-factor that often includes an additional kinetic energy barrier. Main differences in parameterizations of ice nucleation rates concern the way they parameterize the pre-factor and in their assumption of the solid phase that nucleates. While older parameterizations presume the nucleation of hexagonal ice (Ickes et al., 2015; Zobrist et al., 2007; Pruppacher and Klett, 1997), more recent ones assume the formation of stacking disordered or cubic ice (Murray et al., 2010; Koop and Murray, 2016; Němec, 2013; Laksmono et al., 2015), with consequences for the equilibrium vapour pressure over ice.

The pre-factor is usually parameterized in terms of viscosity or the self-diffusion coefficient of liquid water. In both cases, the experimental data range needs to be extrapolated to lower temperatures, usually applying the empirical Vogel-Fulcher-Tammann equation or a power law (Jenniskens and Blake, 1996; Smith and Kay, 1999; Angell, 1995; Koop and Murray, 2016; Murray et al., 2010).

In order to reach agreement with experimentally observed nucleation rates, the interfacial tension between ice and liquid water is generally used as the tuning parameter (Ickes et al., 2015). While there is reasonable agreement between measured nucleation rates for temperatures above 234 K, there is large disagreement at lower temperatures. Nucleation rates covering the temperature range from 238 to 234 K, which are usually measured on micrometre-sized droplets, show discrepancies among each other of up to two orders of magnitude, most probably arising from uncertainties in absolute temperature measurements (Ickes et al., 2015; Riechers et al. 2013). Measurements below 234 K, which require drastically increased cooling rates and/or extremely small sample volumes (Bartell and Chushak, 2005; Manka et al., 2012; Laksmono et al., 2015; Amaya and Wyslouzil, 2018; Kimmel et al., 2019) show systematic discrepancies between each other, which are outside the error range of the different techniques. Therefore, parameterizations need to choose with which datasets they want to comply at low temperatures.

Some parameterizations have a restricted application range. Zobrist et al. (2007) and Pruppacher and Klett (1997) are limited to $T > 230$ K. The parameterization by Koop and Murray (2016) claims to be well constrained by experiments, yet, it predicts critical radii $r_c < 0$ nm for $T < 220$ K. Hence, it is only applicable above 220 K. Given that for ice nucleation within pores, the temperature range below 230 K is most relevant, these parameterizations cannot be used.

To explore the range of predictions for the "no-man's land of ice nucleation", two different parameterizations are compared here, namely the ones by Ickes et al. (2015; Ick15) and Murray et al. (2010, Mr10). Both parameterizations give physically reasonable values over the whole atmospherically relevant temperature range down to 180 K. However, these parameterizations differ in their assumption of the ice phase that nucleates and the treatment of the pre-factor, as will be outlined below. To account for the effect of tension (negative pressure) within the pores, these parameterizations are extended to include pressure dependent

formulations of nucleation rates.

**B1 Parameterization by Ickes et al. (2015)**

The Ick15 parameterization of homogeneous ice nucleation rates has the form:

$$J_{hom} = C_{prefac}\exp\left(\frac{\Delta G_c(T,P_0)}{kT}\right) exp\left(\frac{\Delta F_{diff}(T,P_0)}{kT}\right), \tag{B1}$$

with a constant pre-exponential factor $C_{prefac} = 10^{41}$ m$^{-3}$s$^{-1}$. The thermodynamic energy barrier is formulated in terms of

$\Delta G_c(T,P_0)$, and the diffusion-activation energy of a water molecule to cross the water/ice embryo interface, $\Delta F_{diff}(T, P_0)$, is given as:

$$\Delta F_{diff}(T,P_0) = \frac{\vartheta \ln D(T,P_0)}{\vartheta T}kT^2. \tag{B2}$$

The Ick15 parameterization uses the empirical Vogel-Fulcher-Tammann equation with the parameterization proposed by Smith and Kay (1999) to express the temperature dependence of the water diffusivity:

$$D(T,P_0) = D_0 exp\left(-\frac{E}{T-T_0}\right), \tag{B3}$$

with $D_0 = 3.06 \cdot 10^{-7}$ m$^2$s$^{-1}$, $E = 892$ K and $T_0 = 118$ K valid in the temperature range from 150 to 273 K, yielding

$$\Delta F_{diff}(T,P_0) = \frac{kT^2E}{(T-T_0)^2}. \tag{B4}$$

The Gibbs free energy for the formation of the critical ice embryo, $\Delta G_c(T,P_0)$, is given as:

$$\Delta G_c(T,P_0) = \frac{16\pi\gamma_{iw}(T,P_0)^3 v_i(T,P_0)^2}{3\left(kT\ln\left(\frac{p_w(T,P_0)}{p_{ih}(T,P_0)}\right)\right)^2}. \tag{B5}$$

Here, $p_w(T,P_0)$ and $p_{ih}(T,P_0)$ are the equilibrium vapour pressures of supercooled liquid water and hexagonal ice, respectively, from the parameterization of Murphy and Koop (2005) as reproduced in Eqs. 18 and 19 of the main text. The interfacial tension between ice and liquid water, $\gamma_{iw}(T,P_0)$, is assumed to show a linear temperature dependence and is parameterized as

$$\gamma_{iw}(T,P_0) = 0.03 - 0.18 \cdot 10^{-3}(273.15\ K - T), \tag{B6}$$

where $\gamma_{iw}$ has units of Jm$^{-2}$.

For the formulation of a pressure dependent nucleation rate, the pressure dependence of both, the kinetic and the thermodynamic energy barriers need to be considered by replacing $P_0$ by $P$ in Eq. B1, yielding:

$$J_{hom} = C_{prefac}\exp\left(\frac{\Delta G_c(T,P)}{kT}\right) exp\left(\frac{\Delta F_{diff}(T,P)}{kT}\right). \tag{B7}$$

The pressure dependence of $\Delta G_c(T,P)$ is given by inserting the pressure dependent chemical potential into Eq. B5, yielding:

$$\Delta G_c(T,P) = \frac{16\pi\gamma_{iw}(T,P)^3 v_i(T,P_0)^2}{3\left(kT\ln\left(\frac{p_w(T,P_0)}{p_{ih}(T,P_0)}\right)-(P-P_0)v_i(T,P_0)+(P-P_0)\frac{v_w(T,P)+v_w(T,P_0)}{2}\right)^2}. \tag{B8}$$

Here, the temperature dependence of the molecular volume of ice is again neglected, i.e. $v_i(T,P) = v_i(T,P_0)$. The diffusion-activation energy of a water molecule to cross the water/ice embryo interface depends on the water diffusivity, which is pressure

dependent. The pressure dependence of the self-diffusion coefficient of water has been measured by Prielmeier et al. (1988) in the temperature range from 203.5 – 363 K and for pressures up to 400 MPa. The self-diffusion of water increases by 10 – 70 % along isotherms up to about 100 – 200 MPa and then decreases again when pressure further is increased to 400 MPa. This temperature dependence can be accounted for by introducing a pressure dependent $T_0(P)$ in Eq. B4:

$$T_0(P) = 117.6 - 0.07416P + 0.0002213P^2, \tag{B9}$$

with $P$ given in MPa.

The thermodynamic energy barrier contains the interfacial tension $\gamma_{iw}(T,P)$ as only additional pressure dependent parameter, as the pressure dependence of the molecular volume of ice is neglected. Increased pressure decreases the number of tetrahedral coordinated water molecules and makes water less similar to ice, which should increase the interfacial tension. Indeed, to bring the nucleation rate in agreement with the experimental pressure dependent freezing data given in Fig. 4, the interfacial tension needs to increase with increasing pressure, yielding the following expression:

$$\gamma_{iw}(T,P) = 0.03 - 0.18 \cdot 10^{-3}(273.15\,K - T) + 4.99 \cdot 10^{-5}P - 1.37 \cdot 10^{-7}P^2 + 1.53 \cdot 10^{-10}P^3 + 1.40 \cdot 10^{-12}P^4 -$$
$$2.97 \cdot 10^{-15}P^5 - 3.05 \cdot 10^{-17}P^6, \tag{B10}$$

where $P$ is the absolute pressure in units of MPa and $\gamma_{iw}(T,P)$ the interfacial tension in units of Jm$^{-2}$. The validity range of this parameterization is from 200 to 260 K and from -200 to 160 MPa.

## B2 Parameterization by Murray et al. (2010)

The Murray parameterization is formulated without an exponential term for the kinetic energy barrier:

$$J_{hom} = \frac{2(\gamma_{iw}(T,P_0)kT)^{0.5}}{v_i(T,P_0)^{5/3}\eta(T,P_0)}\exp\left(\frac{\Delta G_c(T,P_0)}{kT}\right). \tag{B11}$$

The pre-factor is taken from Jenniskens and Blake (1996) and shows a dependence on viscosity $\eta(T)$. The temperature dependence of viscosity is parameterized as an adapted Vogel-Fulcher-Tammann relation:

$$\eta(T,P_0) = \eta_0 exp\left(\frac{DT_0}{T-T_0}\right), \tag{B12}$$

with the fragility parameter (Angell, 1995) $D = 10$, $\eta_0 = 10^5$ Pas, and $T_0 = 108.33$ K.

The Gibbs free energy for the formation of the critical ice embryo, $\Delta G_c(T,P_0)$, is given as:

$$\Delta G_c(T,P_0) = \frac{16\pi\gamma_{iw}(T,P_0)^3 v_i(T,P_0)^2}{3\left(kTln\left(\frac{p_w(T,P_0)}{p_{sd}(T,P_0)}\right)\right)^2}. \tag{B13}$$

Here $p_w(T,P_0)$ and $p_{sd}(T,P_0)$ are the equilibrium vapour pressures of supercooled liquid water and stacking disordered ice, respectively, as given in Eqs. 18 – 20 of the main text.

The interfacial tension between ice and water, $\gamma_{iw}(T,P_0)$, is parameterized as:

$$\gamma_{iw}(T,P_0) = 0.0208\left(\frac{T}{235.8\,K}\right)^n, \tag{B13}$$

having units of Jm$^{-2}$. Murray et al. (2010) find that for $n = 0.3$, the parameterization passes best through their experimental data, while $n = 0.97$ is needed to fit the data of Huang and Bartell (1995) at 200 K.

To extend the parameterization to cover both negative and high pressures, the pressure dependence of the molecular volume is neglected, and the thermodynamic energy barrier is modified to:

$$\Delta G_c(T,P) = \frac{16\pi\gamma_{iw}(T,P)^3 v_i(T,P_0)^2}{3\left(kTln\left(\frac{p_w(T,P_0)}{p_{sd}(T,P_0)}\right)-(P-P_0)v_i(T,P_0)+(P-P_0)\frac{v_w(T,P)+v_w(T,P_0)}{2}\right)^2}. \tag{B14}$$

The pressure dependence of the viscosity of water $\eta(T,P)$ has been investigated by Först et al. (2000) within the temperature range from 260 K to 293 K. For supercooled water, viscosity slightly decreases in the pressure range from ambient to about 100 MPa, followed by a slight increase up to 700 MPa. Overall, the variation of viscosity in the investigated pressure and temperature range is less than a factor of two. Since the pre-factor given in Eq. B11, just depends inversely proportional on viscosity,

doubling viscosity decreases the nucleation rate just by a factor of two, which is negligible considering overall uncertainties in measurements and parameterizations. Therefore, the pressure dependence of viscosity is neglected and pressure is assumed to act only on the interfacial tension.

With this assumption, the following formulations of interfacial tensions are obtained by adjusting the calculated nucleation rates to the experimental pressure dependent freezing curve shown in Fig. 4:

For $n = 0.3$

$$\gamma_{iw}(T,P) = 0.0208 \left(\frac{T}{235.8 \, K}\right)^{0.3} + 3.15 \cdot 10^{-5} P - 2.14 \cdot 10^{-7} P^2 + 1.63 \cdot 10^{-10} P^3 + 3.86 \cdot 10^{-12} P^4 - 3.63 \cdot 10^{-15} P^5 -$$
$$9.61 \cdot 10^{-17} P^6. \tag{B15}$$

For $n = 0.97$

$$\gamma_{iw}(T,P) = 0.0208 \left(\frac{T}{235.8 \, K}\right)^{0.97} + 4.14 \cdot 10^{-5} P - 1.69 \cdot 10^{-7} P^2 - 8.01 \cdot 10^{-12} P^3 + 1.41 \cdot 10^{-12} P^4 + 3.10 \cdot 10^{-15} P^5 -$$
$$2.96 \cdot 10^{-17} P^6. \tag{B16}$$

where $P$ is pressure in units of MPa and $\gamma_{iw}(T,P)$ the interfacial tension in units of $Jm^{-2}$. The validity range of these equations is from 200 to 260 K and from -200 to 160 MPa.

**Appendix C: List of symbols**

| | |
|---|---|
| $a$ | extension of the ice embryo along the pore as depicted in Fig. 5 |
| $B, b$ | coefficients to parameterize $\gamma_{vw}(T)$ in Eq. 3 |
| $C_{prefac}$ | pre-exponential factor ($10^{41}$ m$^{-3}$s$^{-1}$) used in the CNT parameterization by Ickes et al. (2015) |
| $D$ | parameter used in the Vogel-Fulcher-Tammann equation |
| $D(T,P_0)$ | temperature dependent water diffusivity at standard pressure $P = 0.1$ MPa |
| $D_0$ | water diffusivity parameter used in the Vogel-Fulcher-Tammann equation |
| $D_p$ | diameter of a cylindrical or conical pore |
| $D_1$ | pore width of a wedge-shaped pore or trench |
| $D_2$ | pore length of a wedge-shaped pore or trench |
| $D_p(T)$ | diameter of pore filling of a cylindrical or conical pore |
| $D_{pfg}(T)$ | pore diameter of cylindrical or conical pores required for free growth of ice out of the pore |
| $D_1(T)$ | diameter of pore filling of a wedge-shaped pore or trench |
| $D_{1fg}(T)$ | pore diameter of trenches or wedges required for free growth of ice out of the pore |
| $E$ | temperature parameter used in the Vogel-Fulcher-Tammann equation |
| $\Delta F_{diff}(T,P_0)$ | temperature dependent diffusion-activation energy of a water molecule to cross the water/ice embryo interface |
| $\Delta F_{diff}(T,P)$ | temperature and pressure dependent diffusion-activation energy of a water molecule to cross the water/ice embryo interface |
| $\Delta G(T,P)$ | temperature and pressure dependent Gibbs free energy to form a spherical ice cluster |
| $\Delta G_c(T,P_0)$ | temperature dependent Gibbs free energy barrier to form ice homogeneously at standard pressure $P_0 = 0.1$ MPa |
| $\Delta G_c(T,P)$ | temperature and pressure dependent Gibbs free energy barrier to form ice homogeneously |
| $\Delta G_{gr}(T,P)$ | Gibbs free energy to grow a spherical cap on top of a pore |
| $h$ | height of the spherical ice cap as depicted in Fig. 10 |
| $J_{hom}$ | homogeneous ice nucleation rate |
| $k$ | Boltzmann constant |
| $M_w$ | molecular mass of water |

| | | |
|---|---|---|
| $n$ | | exponential parameter used in the CNT parameterization by Murray et al. (2010) |
| $N_a$ | | Avogadro constant |
| $p$ | | equilibrium vapour pressure above the (curved) water surface |
| $p_w(T,P_0)$ | | temperature dependent equilibrium vapour pressure of water at standard pressure $P_0 = 0.1$ MPa |
| 5 | $p_w(T,P)$ | temperature and pressure dependent equilibrium vapour pressure of water |
| $p_{ih}(T,P_0)$ | | temperature dependent equilibrium vapour pressure of hexagonal ice at standard pressure $P_0$ |
| $p_{sd}(T,P_0)$ | | temperature dependent equilibrium vapour pressure of stacking disordered ice at standard pressure $P_0$ |
| $P$ | | absolute pressure in MPa |
| $P_0$ | | standard pressure (0.1 MPa) |
| 10 | $\Delta P$ | pressure difference across the vapour/water interface of curved surfaces |
| $r$ | | radius of the emerging ice embryo |
| $r_c(T,P)$ | | critical radius of the ice embryo |
| $r_s(T,P)$ | | radius of the ice embryo when $\Delta G(T, P) = 0$ |
| $r_m(T)$ | | radius of the curved water surface of cylindrical or conical pores |
| 15 | $r_{op}(T)$ | radius of the pore opening as depicted in Fig. 10 |
| $r_1(T), r_2(T)$ | | principal radii of curvature of the water surface in wedge-shaped pores or trenches as explained in Fig. 2 |
| $R$ | | universal gas constant |
| $S_c$ | | critical water saturation ratio for pore filling |
| $S_i$ | | ice saturation ratio of the gas phase |
| 20 | $S_w$ | water saturation ratio of the gas phase |
| $t$ | | thickness of quasi-liquid layer (QLL) |
| $T$ | | absolute temperature in Kelvin |
| $T_0$ | | temperature parameter used in the Vogel-Fulcher-Tammann equation |
| $T_0(P)$ | | pressure dependent temperature parameter used in the Vogel-Fulcher-Tammann equation |
| 25 | $T_c$ | critical temperature of water (647.096 K) used to parameterize $\gamma_{vw}(T)$ (Eq. 3) |
| $x$ | | radius increase of the base of the spherical cap beyond the pore radius as depicted in Fig. 10 |
| $\gamma(T,P)$ | | temperature and pressure dependent interfacial tension |
| $\gamma_{vw}(T)$ | | surface tension of the vapour/water interface |
| $\gamma_{vi}(T)$ | | surface tension of the vapour/ice interface |
| 30 | $\gamma_{is}(T)$ | interfacial tension between ice and the outer surface surrounding the pore |
| $\gamma_{iw}(T,P)$ | | interfacial tension between ice and water |
| $\delta$ | | Tolman length |
| $\eta(T)$ | | viscosity |
| $\eta_0(T)$ | | viscosity parameter used in the Vogel-Fulcher-Tammann equation |
| 35 | $\theta_{ws}$ | contact angle of water ($w$) on the pore surface ($s$) |
| $\theta_{is}(T)$ | | contact angle of ice ($i$) on the pore surface ($s$) |
| $\theta_{iw}(T)$ | | contact angle between ice ($i$) and water ($w$) |
| $\kappa(T)$ | | compressibility of liquid water |
| $\mu$ | | universal critical exponent (1.256) used to parameterize $\gamma_{vw}(T)$ (Eq. 3) |
| 40 | $\mu_i(T,P_0)$ | temperature dependent chemical potential of ice at standard pressure $P_0$ |
| $\mu_w(T,P_0)$ | | temperature dependent chemical potential of liquid water at standard pressure $P_0$ |
| $\mu_v(T,P_0)$ | | temperature dependent chemical potential of water vapour at standard pressure $P_0$ |

$\mu_i(T,P)$          temperature and pressure dependent chemical potential of ice

$\mu_w(T,P)$          temperature and pressure dependent chemical potential of liquid water

$\mu_v(T,P)$          temperature and pressure dependent chemical potential of water vapour

$\Delta\mu_{iw}$          difference between chemical potentials of ice and liquid water $(\mu_i(T,P) - \mu_w(T,P))$

5   $v_w(T,P_0)$          temperature dependent molecular volume of liquid water at standard pressure

$v_i(T,P_0)$          temperature dependent molecular volume of water in the ice phase at standard pressure

$v_w(T,P)$          temperature and pressure dependent molecular volume of liquid water

$v_i(T,P)$          temperature and pressure dependent molecular volume of water in the ice phase

$\rho_w(T,P_0)$          temperature dependent density of liquid water at standard pressure $P_0$

10   $\rho_w(T,P)$          temperature and pressure dependent density of liquid water

$\tau$          dimensionless distance from the critical temperature of water used to parameterize $\gamma_{vw}(T)$ (Eq. 3)

**Appendix D: Values recommended for checking computer codes**

Table D1 Selected values of density $\rho_w(T,P)$ in (kgm$^{-3}$) parameterized in Eqs. A1 – A5

| T (K)/ P (MPa) | 399 | 200 | 100 | 50 | 0.1 | 0.0* | -20 | -50 | -100 |
|---|---|---|---|---|---|---|---|---|---|
| 330 | 1123.16 | 1066.83 | 1029.06 | 1007.81 | 985.03 | 984.98 | 975.40 | 960.56 | 934.57 |
| 298 | 1121.22 | 1069.84 | 1035.79 | 1016.70 | 996.27 | 996.23 | 987.66 | 974.39 | 951.17 |
| 273 | 1135.44 | 1083.87 | 1046.55 | 1025.03 | 1001.65 | 1001.60 | 991.70 | 976.27 | 949.03 |
| 230 | 1155.14 | 1094.92 | 1039.32 | 1005.17 | 966.86 | 966.78 | 950.23 | 924.15 | 877.29 |
| 210 | 1173.94 | 1106.10 | 1036.55 | 992.88 | 943.38 | 943.28 | 921.78 | 887.75 | 826.29 |

*corresponds to Eq. A1

Table D2 Selected values of the CNT parameterizations by Ickes et al. (2015) and Murray et al. (2010)

| T (K) | P (MPa) | Ickes et al. (2015) | | | Murray et al. (2010), $n = 0.3$ | | | Murray et al. (2010), $n = 0.97$ | | |
|---|---|---|---|---|---|---|---|---|---|---|
| | | $\gamma_{iw}$ (Jm$^{-2}$) | $\Delta G_c$ (J) | $J_{hom}$(Jcm$^{-3}$s$^{-1}$) | $\gamma_{iw}$ (Jm$^{-2}$) | $\Delta G_c$ (J) | $J_{hom}$(Jcm$^{-3}$s$^{-1}$) | $\gamma_{iw}$ (Jm$^{-2}$) | $\Delta G_c$ (J) | $J_{hom}$(Jcm$^{-3}$s$^{-1}$) |
| 235 | 0.1 | 0.023165 | $1.5210 \cdot 10^{-19}$ | $1.0870 \cdot 10^{8}$ | 0.020782 | $1.7763 \cdot 10^{-19}$ | $1.6134 \cdot 10^{8}$ | 0.020736 | $1.7644 \cdot 10^{-19}$ | $2.3212 \cdot 10^{8}$ |
| 235 | 50 | 0.025339 | $2.4197 \cdot 10^{-19}$ | $2.2253 \cdot 10^{-4}$ | 0.021861 | $2.6497 \cdot 10^{-19}$ | $3.3744 \cdot 10^{-4}$ | 0.022387 | $2.8458 \cdot 10^{-19}$ | $8.0831 \cdot 10^{-7}$ |
| 235 | -50 | 0.020313 | $9.4265 \cdot 10^{-20}$ | $1.8630 \cdot 10^{15}$ | 0.018672 | $1.1618 \cdot 10^{-19}$ | $2.5693 \cdot 10^{16}$ | 0.018247 | $1.0843 \cdot 10^{-19}$ | $2.7677 \cdot 10^{17}$ |
| 230 | 0.1 | 0.022265 | $1.1184 \cdot 10^{-19}$ | $4.4925 \cdot 10^{12}$ | 0.020648 | $1.3767 \cdot 10^{-19}$ | $9.9317 \cdot 10^{12}$ | 0.020308 | $1.3097 \cdot 10^{-19}$ | $8.1276 \cdot 10^{13}$ |
| 230 | 50 | 0.024439 | $1.7398 \cdot 10^{-19}$ | $3.4125 \cdot 10^{4}$ | 0.021727 | $1.9645 \cdot 10^{-19}$ | $9.3273 \cdot 10^{4}$ | 0.021959 | $2.0282 \cdot 10^{-19}$ | $1.2618 \cdot 10^{4}$ |
| 230 | -50 | 0.019413 | $7.0050 \cdot 10^{-20}$ | $6.3169 \cdot 10^{17}$ | 0.018539 | $9.2978 \cdot 10^{-20}$ | $1.2185 \cdot 10^{19}$ | 0.017819 | $8.2573 \cdot 10^{-20}$ | $3.1651 \cdot 10^{20}$ |
| 210 | 0.1 | 0.018665 | $4.2656 \cdot 10^{-20}$ | $1.2123 \cdot 10^{19}$ | 0.020093 | $7.5259 \cdot 10^{-20}$ | $5.8364 \cdot 10^{19}$ | 0.018593 | $5.9634 \cdot 10^{-20}$ | $1.2296 \cdot 10^{22}$ |
| 210 | 50 | 0.020839 | $6.4854 \cdot 10^{-20}$ | $2.3765 \cdot 10^{16}$ | 0.021171 | $9.8304 \cdot 10^{-20}$ | $2.1162 \cdot 10^{16}$ | 0.020244 | $8.5952 \cdot 10^{-20}$ | $1.4657 \cdot 10^{18}$ |
| 210 | -50 | 0.015813 | $2.6530 \cdot 10^{-20}$ | $3.5809 \cdot 10^{20}$ | 0.017983 | $5.5167 \cdot 10^{-20}$ | $5.6439 \cdot 10^{22}$ | 0.016105 | $3.9624 \cdot 10^{-20}$ | $1.1370 \cdot 10^{25}$ |

*Data availability.* There is no new data involved.

*Author contributions.* Everything is contributed by CM.

*Competing interests.* The author declares that she has no conflict of interest.

**Acknowledgments**

The author would like to thank Robert O. David, Zamin A. Kanji and Fabian Mahrt for carefully reading and correcting the
manuscript and Beiping Luo for helpful discussions.

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

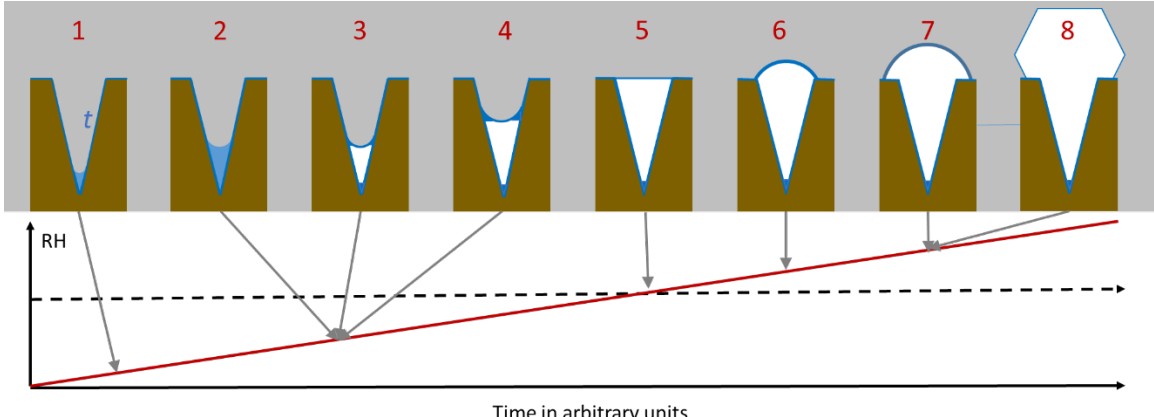

**Figure 1.** Pore condensation and freezing in conical or wedge-shaped pores (brown) assuming continuously increasing RH (red line). The pore is assumed to be covered with a QLL of width $t$ (darker blue) already at low RH. Free water (lighter blue) collects in pores and freezes to ice (white) that further grows within the pore. At ice saturation (dashed black horizontal line), the pore is completely filled with ice.

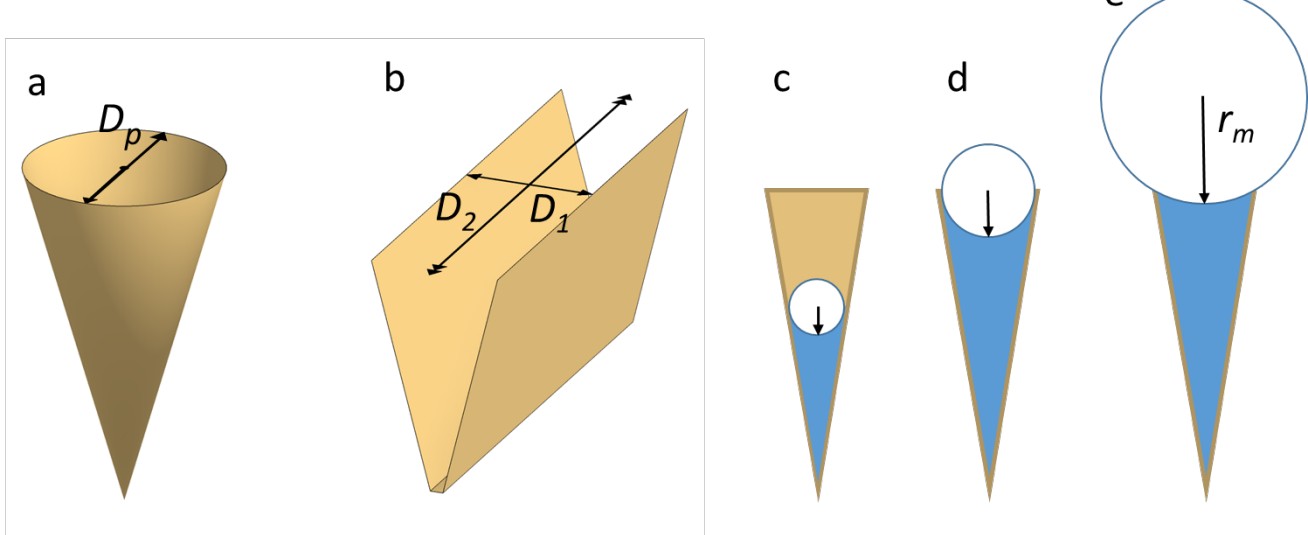

**Figure 2.** Illustration of pore shapes and pore filling: panel (a) shows an empty conical pore with diameter $D_p$ and panel (b) an empty wedge-shaped pore with a width $D_1$ and a length $D_2 = \infty$. Panels (c) to (e) show pore condensation with increasing radius of the meniscus for a conical or wedge-shaped pore assuming complete wetting ($\theta_{ws} = 0°$). The radius of meniscus in a conical or cylindrical pore is denoted $r_m$ and the radius of meniscus in a wedge-shaped pore or trench is denoted $r_1 = D_1/2$ while $r_2 = D_2 = \infty$ (not shown).

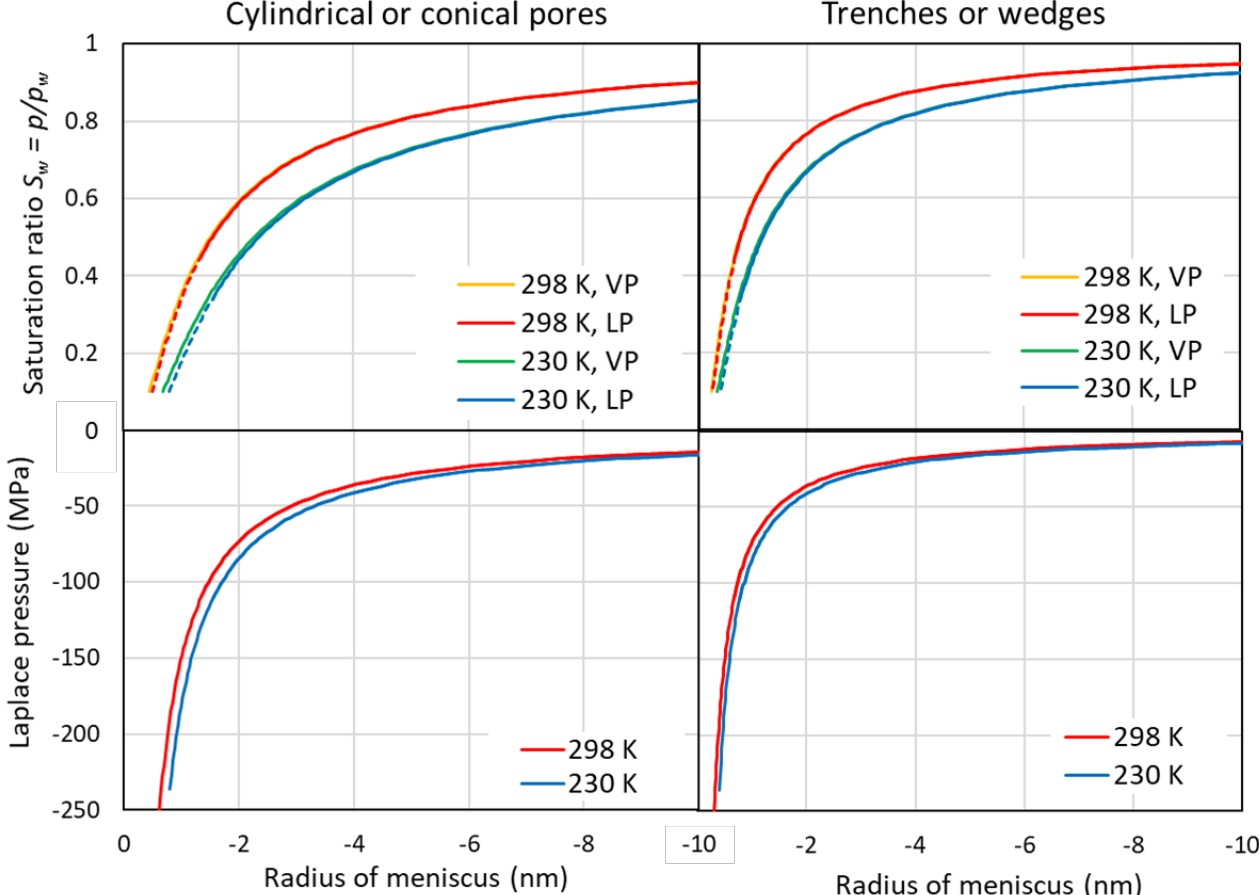

**Figure 3.** Upper panels: Saturation ratio as a function of the meniscus radius of cylindrical or conical pores ($r_m$, left panels) and trenches or wedges ($r_l$, right panels) at 298 K and 230 K. Saturation ratios are given neglecting the effect of negative pressure on the molecular volume of water (indicated as VP in the legend and calculated using Eqs. 1 and 2) and accounting for the Laplace pressure exerted on the pore water (indicated as LP and calculated using Eqs. 8 and 10). The dashed portions of the red and blue lines indicate the extrapolation of the molecular volume to strongly negative pressures as shown in the lower panels. Lower panels: Laplace pressure as a function of the radius of the meniscus.

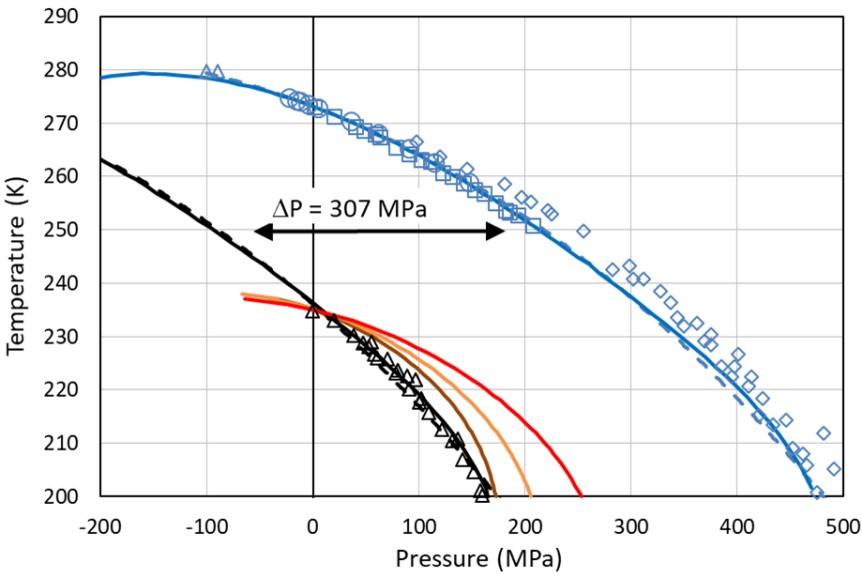

**Figure 4.** Pressure dependence of melting (blue) and freezing (black) temperatures of ice I. Melting point measurements of ice I are from Kanno et al. (1975) (blue squares), Mishima (1996) (blue diamonds), Henderson and Speedy (1987) (blue circles) and Roedder (1967) (blue triangles). For simplicity, melting data of other ice polymorphs are not shown. The blue solid line is the melting curve calculated by setting Eq. 24 to zero. The dashed blue line is a fit to the measured melting temperatures (blue symbols) using the equation given in Marcolli (2017b): $T(K) = 557.2 - 273\exp((300 + P(MPa))^2/2270000)$. Freezing temperatures of ice I (black triangles) are from Kanno et al. (1975). The black solid and dashed lines represent a homogeneous ice nucleation rate of $10^8$ cm$^{-3}$s$^{-1}$ obtained by shifting the blue curves by $\Delta P = 307$ MPa to lower values. Red line: Ick15 parameterization with pressure dependent chemical potential ($\Delta\mu_{iw}$ of Eq. 24), all other parameters without pressure dependence. Orange and brown lines: Mr10 parameterization with pressure dependent chemical potential ($\Delta\mu_{iw}$ of Eq. 24) with $n = 0.3$ (brown) and $n = 0.97$ (orange), all other parameters without pressure dependence. Fully pressure dependent Ick15 and Mr10 parameterizations were optimized to overlay with the experimental freezing data and are not shown here.

**Figure 5.** Ice nucleation and growth within a cylindrical pore of diameter $D_p = 2r + 2t$, with $a$ describing the growth along the pore axis, $t$ being the thickness of the QLL and $r$ being the maximum radius the embryo can reach perpendicular to the pore axis.

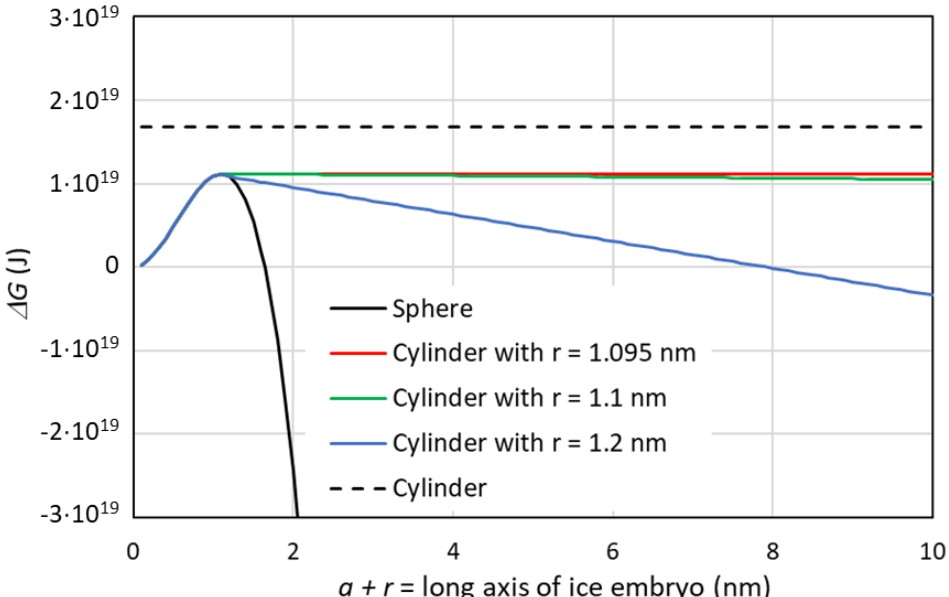

**Figure 6**. Gibbs free energy of a growing ice embryo calculated with CNT using the Ick15 parameterization for $T = 230$ K and $P_0 = 0.1$ MPa. The black solid line describes spherical growth of the ice embryo as expected in bulk water. The black dashed line gives $\Delta G$ for the growth in length of a cylinder starting from a thin disk with radius $r_c = 1.095$ nm. The colored lines show $\Delta G$ for growth in cylindrical pores with radii available for free water of 1.095 nm (critical value, red line), 1.1 nm (green line), and 1.2 nm (blue line), respectively.

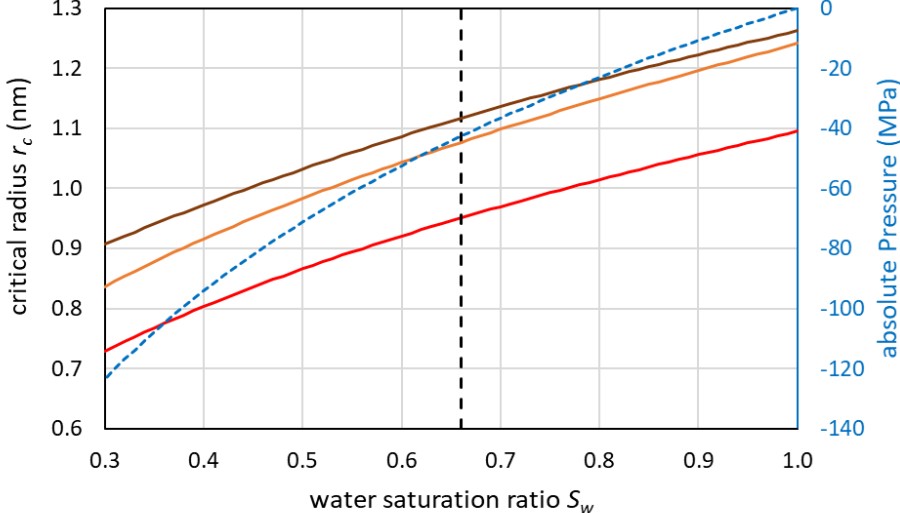

**Figure 7.** Dependence of the critical embryo radius on the water saturation ratio at 230 K for ice nucleation within pores using Ick15 (red) and Mr10 (orange: $n = 0.97$; brown: $n = 0.3$) parameterizations. Saturation with respect to bulk ice is indicated as the black dashed vertical line. The blue dashed line shows the negative pressure that builds up in pore water when the water saturation ratio decreases.

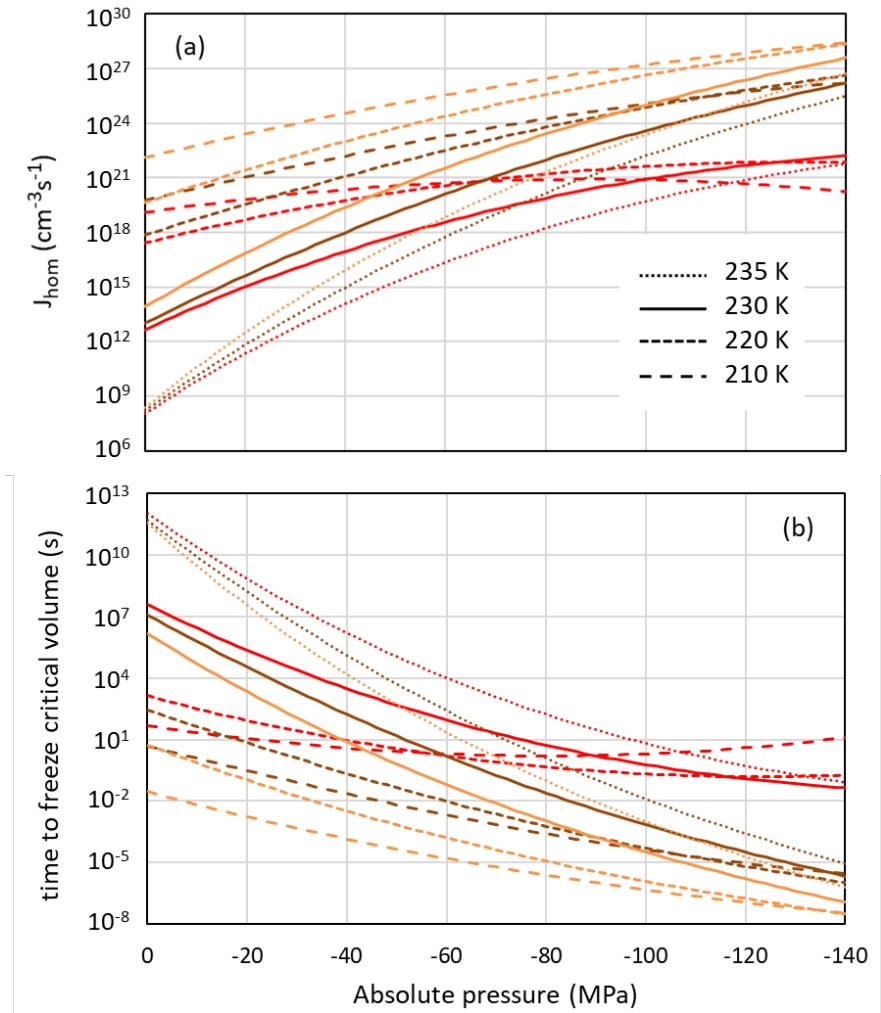

**Figure 8.** Pressure dependence of homogeneous nucleation rates (panel a) and the time to freeze the critical water volume for homogeneous ice nucleation (panel b) at four different temperatures as indicated in the legend. Ick15 is shown in red, Mr10 in orange for $n = 0.97$ and in brown for $n = 0.3$.

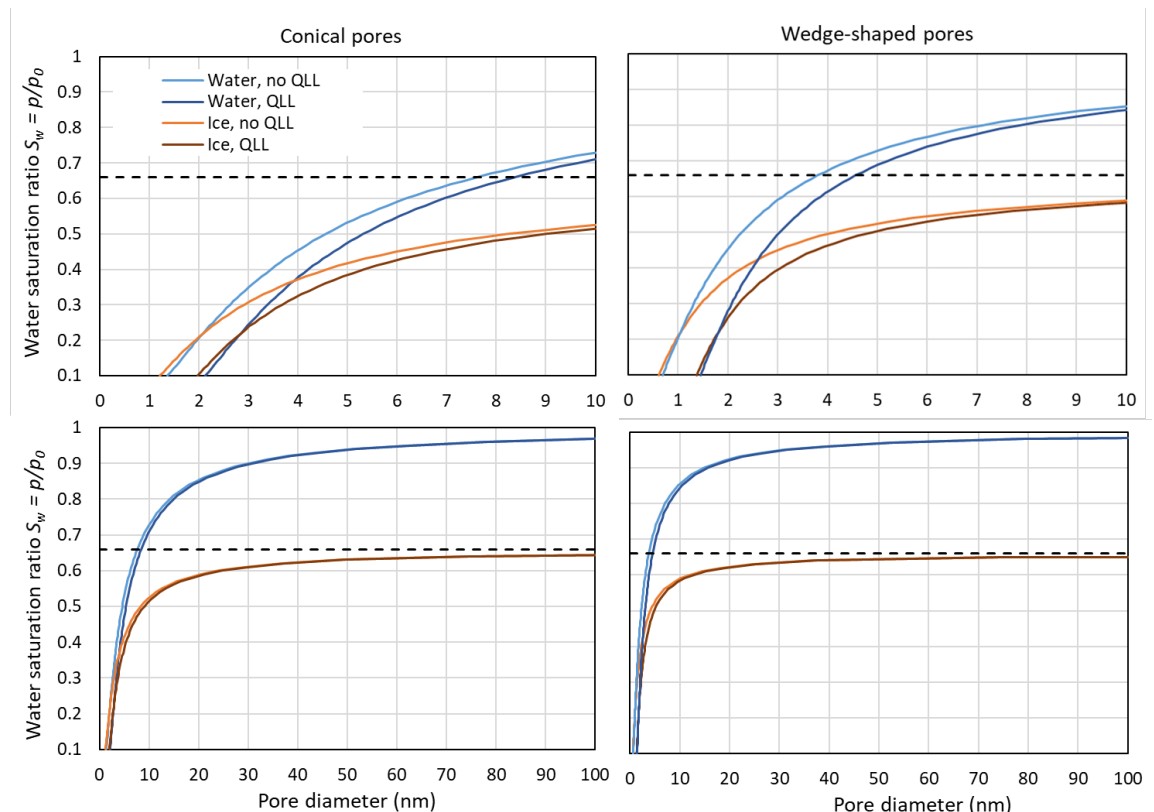

**Figure 9.** Comparison of pore filling with water and ice for conical (left panels) and wedge-shaped pores (right panels) at 230 K. For water and ice, the Kelvin effect is calculated without QLL, given as light blue and light brown lines, respectively, and assuming a QLL with width $t$ = 0.38 nm, given as dark blue and dark brown lines, respectively. The black, dashed horizontal line indicates ice saturation.

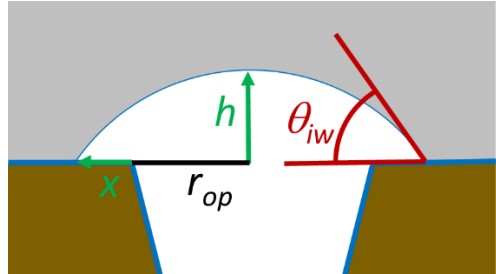

**Figure 10.** Growth of ice out of the pore as a spherical cap with an increasing height $h$ covering an increasing area $\pi(r_{op} + x)^2$.

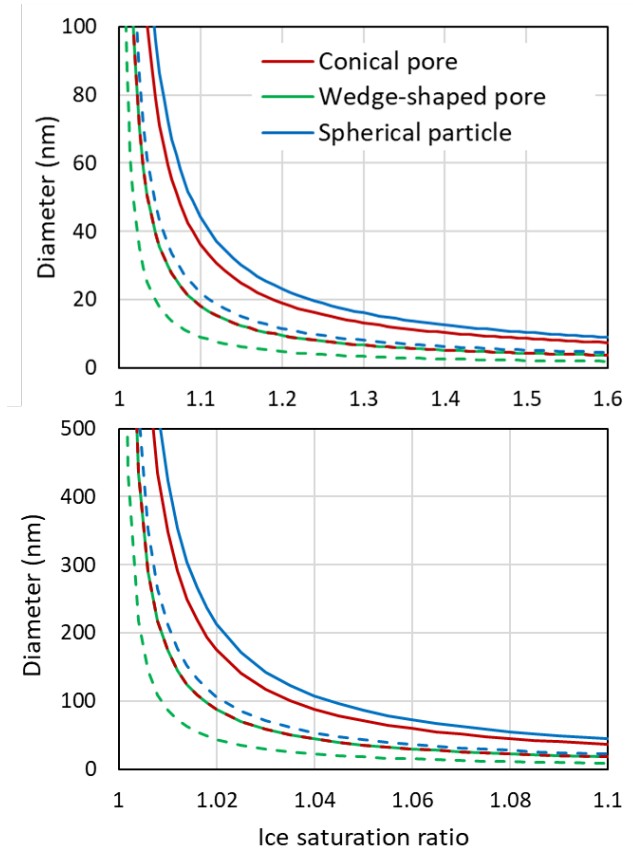

**Figure 11.** Diameters of pore opening allowing barrier-free ice growth out of cylindrical or conical pores (red), wedges or trenches (green), and spherical particles (blue) as a function of ice saturation ratio. The lower panel is a zoomed in view of the upper panel. Solid lines are obtained for $T = 230$ K with $\gamma_{vi} = \gamma_{vw} + \gamma_{iw} = 0.0811$ Jm$^{-2}$ + 0.0226 Jm$^{-2}$ = 0.1033 Jm$^{-2}$, $\gamma_{iw}$ from Ick15, and $\theta_{iw} = 55°$. Dashed lines are calculated by halving all involved interfacial tensions to account for the presence of trace amounts of organic substances that adsorb on surfaces and interfaces. Note that the dashed red line overlays the green solid line.