# Peer review of "Technical note: Fundamental aspects of ice nucleation via pore condensation and freezing including Laplace pressure and growth into macroscopic ice"

_Atmospheric Chemistry and Physics, 2019_

## Referee Comment (RC1) · Anonymous Referee #2 · 1 Dec 2019

I support publication. This technical note collects and organizes a variety of topics necessary to evaluate the pore condensation freezing mechanism in a wide array of circumstances.

I have a few suggestions that might improve the paper around the edges, but the paper could be published "as-is" and still be a solid contribution to the literature.

There are a lot of parameterizations in this paper, and many of them are excruciatingly sensitive. For example, Equation 18 has a term in it with six significant figures. It's

very easy to leave a digit out or transpose a couple when entering expressions like this into code. It would be quite helpful if a table of one or two pair of representative values from equations like this were included in the text. As an example of something like this, see appendix C in Murphy and Koop, QJRMS, 2005, "Review of the vapor pressures..." Equations A1, B15, and B16 are other examples that I think would benefit from something like this. (That's not an exhaustive list.)

**Minor comments**

Line 24: "Cirrus may form through different mechanisms." This sentence seems redundant, considering the fact that the previous sentence was a description of various mechanisms by which cirrus can form.

line 13: "...pore with thickness t = 0.38 - 0.6 nm" That dash looks like a minus sign at first glance. Perhaps replace with "...pore with thickness of 0.38 to 0.6 nm"
* * *

---

## Referee Comment (RC2) · Anonymous Referee #1 · 3 Jan 2020

This is a good paper and correctly categorized as a technical note as it does not contain considerable new scientific findings. It follows on several papers, including several by Marcolli et al., and the advance here is that it provides a more comprehensive treatment of the theory of pore condensation freezing than in previous works. As a technical note it represents an important resource and should be highly referenced by those working on this ice nucleation mechanism. It should be published with minor changes.

Some suggestions for the author to consider:

[Figure]

1. The introduction only discusses cirrus clouds as the application of pore condensation freezing. The author may consider if other systems might be impacted, e.g. those beyond the atmosphere. 2. Also in the introduction, can it be stated that homogenous freezing via pore condensation does not exclude heterogeneous mechanisms? The term 'prevailing mechanism' in line 28/29 makes it unclear if the author is suggesting this is the dominant mechanism (which I do not believe has been shown) or one of several depending on specific conditions (which I believe is the consensus). A sentence or two to clarify would be helpful. 3. Given the density of equations in Section 2 and after, it would be helpful to have a table with variable definitions in the paper, e.g., Appendix A and then use B for the derivations / parameterizations. 4. The use of Figures 1 and 11 is appreciated for clarify on the specifics of pore modeling, this will be of help to the broader readership of this Technical Note. The author may consider moving Figure 11 earlier in the paper as a description of how this mechanism is impactful on the atmosphere. 5. I concur with the point made by Reviewer 2 regarding significant figures, e.g. Section 3.1.1 in the equations and perhaps this could be incorporated into a table or another Appendix.

---

## Author Comment (AC1) · 7 Feb 2020

**Responses to Anonymous Referee #1**

*I thank the reviewer for his/her constructive comments that I address below point by point (responses are in italic).*

This is a good paper and correctly categorized as a technical note as it does not contain considerable new scientific findings. It follows on several papers, including several by Marcolli et al., and the advance here is that it provides a more comprehensive treatment of the theory of pore condensation freezing than in previous works. As a technical note it represents an important resource and should be highly referenced by those working on this ice nucleation mechanism. It should be published with minor changes.

Some suggestions for the author to consider:

1. The introduction only discusses cirrus clouds as the application of pore condensation freezing. The author may consider if other systems might be impacted, e.g. those beyond the atmosphere.

*Indeed, PCF is applicable also in other fields. Yet, since this technical note will be published in an atmospheric journal, I would like to keep the focus on the atmosphere.*

2. Also in the introduction, can it be stated that homogenous freezing via pore condensation does not exclude heterogeneous mechanisms? The term 'prevailing mechanism' in line 28/29 makes it unclear if the author is suggesting this is the dominant mechanism (which I do not believe has been shown) or one of several depending on specific conditions (which I believe is the consensus). A sentence or two to clarify would be helpful.

*This sentence refers to Marcolli (2014), where it is indeed claimed that PCF is the prevailing nucleation mechanism at low ice supersaturation (i. e. below water saturation in the mixed-phase cloud regime, and below homogeneous freezing of solution droplets at cirrus conditions). Further down in the introduction, evidence supporting this statement is given. Conversely, there is indeed no experimental evidence that substantiates a depositional ice nucleation mechanism, i.e. is able to prove the absence of liquid water in deposition nucleation.*

3. Given the density of equations in Section 2 and after, it would be helpful to have a table with variable definitions in the paper, e.g., Appendix A and then use B for the derivations / parameterizations.

*I implemented a table with variable definitions as Appendix C to the paper.*

4. The use of Figures 1 and 11 is appreciated for clarify on the specifics of pore modeling, this will be of help to the broader readership of this Technical Note. The author may consider moving Figure 11 earlier in the paper as a description of how this mechanism is impactful on the atmosphere.

*Thank you for this suggestion. This figure is now Fig. 1 of the revised manuscript in the new Sect. 2 (Atmospheric scenario of PCF).*

5. I concur with the point made by Reviewer 2 regarding significant figures, e.g. Section 3.1.1 in the equations and perhaps this could be incorporated into a table or another Appendix.

*I added Appendix D to the revised manuscript with Table D1 listing values of the density parameterization given in Appendix A and Table D2 giving values of the pressure dependent CNT parameterizations described in Appendix B.*

---

## Author Comment (AC2) · 7 Feb 2020

**Responses to Anonymous Referee #2**

*I thank the reviewer for his/her constructive comments that I address below point by point (responses are in italic).*

I support publication. This technical note collects and organizes a variety of topics necessary to evaluate the pore condensation freezing mechanism in a wide array of circumstances. I have a few suggestions that might improve the paper around the edges, but the paper could be published "as-is" and still be a solid contribution to the literature.

There are a lot of parameterizations in this paper, and many of them are excruciatingly sensitive. For example, Equation 18 has a term in it with six significant figures. It's very easy to leave a digit out or transpose a couple when entering expressions like this into code. It would be quite helpful if a table of one or two pair of representative values from equations like this were included in the text. As an example of something like this, see appendix C in Murphy and Koop, QJRMS, 2005, "Review of the vapor pressures..." Equations A1, B15, and B16 are other examples that I think would benefit from something like this. (That's not an exhaustive list.)

*I added Appendix D to the revised manuscript with Table D1 listing values of the density parameterization given in Appendix A and Table D2 giving values of the pressure dependent CNT parameterizations described in Appendix B. I did not include a table listing values of Eq. 18 as this parameterization is taken from Murphy and Koop (2005) and values are given therein.*

**Minor comments**

Line 24: "Cirrus may form through different mechanisms." This sentence seems redundant, considering the fact that the previous sentence was a description of various mechanisms by which cirrus can form.

*I agree. I removed this sentence in the revised manuscript.*

line 13: "...pore with thickness t = 0.38 - 0.6 nm" That dash looks like a minus sign at first glance. Perhaps replace with "...pore with thickness of 0.38 to 0.6 nm"

*I changed the manuscript accordingly.*

---

## Author Response (AR2)

**Editor Decision: Publish subject to technical corrections** (10 Feb 2020) by Barbara Ervens
Comments to the Author:
Dear Claudia,

thank you for addressing all reviewer comments and providing a carefully revised manuscript. I'm happy to accept it for publication in ACP.
I have several suggestions for minor (technical) corrections below. Please take them into account prior to uploading the final files or during proofreading.

Best regards
Barbara
* * *
p. 5, l.3/ 4: can be shortened to '…depends on T and P' as they have been defined before
p. 5, l. 16: (P0 = 0.1 MPa) can be removed
p. 6, l. 12: should this read '…are given in Appendix A1 as Eq. A1 and Eqs A2 – A6, respectively.' (not '…are given in Appendix A1 as Eq. A1 and Eqs A1 – A5, respectively.')
p. 7, l. 18-21 ('Assuming…'): This is a very long sentence. Please split into two.
p. 7, l. 39: 'The green line assumes….'. Please reword, e.g. 'It is assumed that the ice embryo has a spherical shape of r = 1.1 nm when it reaches the QLL and then starts to grow in length as a cylinder with half spheres at its ends (green line in Fig. 6).'
p. 7, l. 41: replace 'requesting' by 'requiring'
p. 8, l. 5: replace 'requested' by 'required'
p. 8, l. 19: either 'melts' or 'may melt'
p. 10, l. 21: replace 'realizes' by 'reaches' (check meaning)
p. 10, l. 31: add degree symbol (0°)
p. 12, l. 12: write units consistently, i.e. kg m-3
p. 15, l. 37: you can write either 'depends inversely on…' or 'is inversely proportional…'
p. 17, l. 1: 'exponential parameter' does not seem a common term. Why not simply 'exponent'?
p. 25, l. 3: QLL has not been defined before

*Dear Barbara*

*Thank you very much for carefully reading through my manuscript and correcting it. I revised the manuscript accordingly. See below.*

*Best regards*
*Claudia*

[revised manuscript text omitted]